# Mapping cellular-scale internal mechanics in 3D tissues with thermally responsive hydrogel probes

Stephanie Mok [1], Sara Al Habyan[2], Charles Ledoux [1], Wontae Lee [1], Katherine N. MacDonald[1], Luke McCaffrey[2] & Christopher Moraes [1,2,3✉]

Local tissue mechanics play a critical role in cell function, but measuring these properties at cellular length scales in living 3D tissues can present considerable challenges. Here we present thermoresponsive, smart material microgels that can be dispersed or injected into tissues and optically assayed to measure residual tissue elasticity after creep over several weeks. We first develop and characterize the sensors, and demonstrate that internal mechanical profiles of live multicellular spheroids can be mapped at high resolutions to reveal broad ranges of rigidity within the tissues, which vary with subtle differences in spheroid aggregation method. We then show that small sites of unexpectedly high rigidity develop in invasive breast cancer spheroids, and in an in vivo mouse model of breast cancer progression. These focal sites of increased intratumoral rigidity suggest new possibilities for how early mechanical cues that drive cancer cells towards invasion might arise within the evolving tumor microenvironment.

[1] Department of Chemical Engineering, McGill University, 3610 University Street, Montreal, QC H3A 0C5, Canada. [2] Rosalind and Morris Goodman Cancer Research Centre, McGill University, 160 Pine Ave W, Montreal, QC H3A 1A3, Canada. [3] Department of Biomedical Engineering, McGill University, 3775 University Street, Montreal, QC H3A 2B4, Canada. ✉email: chris.moraes@mcgill.ca

Exquisitely structured tissues and organs arise from a homogenous blastomere through spatial patterns of cell proliferation, migration, and differentiation, in concert with matrix secretion and remodeling[1–3]. Mechanical features of the local microenvironment are critical regulators of these cellular processes[4–11], and tissue stiffness is now well-established to drive fate-function relationships during development[12,13]; disease progression[14–17] and tissue homeostasis[18–20]. However, our technical ability to monitor and characterize tissue mechanics at the cellular length scale during tissue development remains severely limited, and could be critically important in elucidating biophysical mechanisms of tissue morphogenesis and disease.

Conventional mechanical characterization techniques provide only a limited view of tissue rigidity, particularly at the meso-length scale of individual cells. Macroscale measurement tools such as tensional or shear rheometry cannot capture local mechanical variations around cells[21], while high-resolution tools such as atomic force microscopy are ideally suited for sub-cellular nanoscale measurements, and are limited to measuring near-surface stiffness in two-dimensional or cut tissue sections. Although non-contact techniques such as ultrasound elastography or magnetic cytometry[22–24] provide limited remote access to address these issues of scale, they cannot mimic a cell's ability to interrogate the surrounding tissue by applied deformations with stroke lengths of 10 s of microns[25,26].

Serwane et al. recently developed an intriguing strategy to measure tissue mechanics with injectable, cell-sized, magnetic oil droplets, that deform in response to applied magnetic fields to quantify local tissue mechanics in soft tissues such as zebrafish embryos[27]. This powerful approach provides unique insight into highly local evolution of tissue mechanics during development, but the small droplet volumes allow only very low magnetic actuation forces, limited stroke lengths, and can only measure stiffnesses of <1 kPa. Moreover, oil droplets also split apart during large scale morphogenesis, limiting the monitoring period. Finally, this technique requires specialized equipment and expertise for simultaneous magnetic and optical probing, which limits experiments to small, thin, transparent, tissues that can stimulated with a uniform magnetic field. To circumvent some of these limitations, we build upon recent materials-based strategies using swellable hydrogels to generate local deformations within porous materials[21].

Here, thermoresponsive hydrogel microspheres, termed microscale temperature-actuated mechanosensors (µTAMs) are used to measure a wide range of residual tissue elasticities within 3D biomaterials, at the length-scales of individual cells, in engineered tissues or animal models. µTAMs are spherical, thermoresponsive hydrogels that remain compact at tissue culture temperatures, but swell when cooled by a few degrees. By measuring the degree to which they expand, the residual elasticity after creep of the surrounding tissue can be inferred (Fig. 1a). In this work, we first develop the design principles to optimize hydrogel formulations for soft tissue measurements; and then demonstrate that µTAMs can be integrated into engineered tissues and animal models. These studies reveal that significantly different internal residual elasticities arise in multicellular aggregates based on the aggregation method; and that highly localized hot spots of considerably elevated intratumoral rigidity emerge during establishment of a metastatic breast tumor.

## Results

### Design and characterization of µTAMs.
Poly N-isopropylacrylamide (PNiPAAM) hydrogels are tunable, biocompatible, thermoresponsive materials that remain compact at 37 °C, but reversibly swell at slightly lower temperatures when solute interactions favor hydrophilic domains of the polymer[28,29]. To form PNiPAAM gels into µTAM probes, microspherical droplets of hydrogel formulations were polymerized with a fluorescent label[30], in an oxygen-free, oil/water vortex emulsion (Fig. 1b). This produces polydisperse hydrogel particles with expanded diameters that range from 10 to 100 µm (Supplementary Fig. 1), which is comparable to the size and mechanical sensing range of many adherent cells when compact[31]. The fabricated µTAMs retain their ability to reversibly shrink above a lower critical solution temperature of ~34 °C[32] (Fig. 1c, d). The thermoresponsive diameter change was independent of µTAM size, and tunable based on the hydrogel formulation (Supplementary Fig. 2). Free expansion in solution was tunable between $1.92 \pm 0.05$ and $3.4 \pm 0.18$ for the polymer formulations tested. To confirm suitability in tissue culture conditions, we tested µTAM expansion in physiologic protein-rich conditions, as long-chain molecules in the cellular milieu may molecularly crowd and interfere with the polymer-water interactions necessary for expansion. Free expansion ratios of non-functionalized µTAMs were not significantly altered in even 100% fetal bovine serum (FBS; Supplementary Fig. 2), which contains supraphysiologically high levels of soluble protein[33].

µTAMs require an adhesive matrix protein coating to support integration into tissues, which may impact their expansion characteristics through transport limitations or mechanical restrictions. Collagen I was selected as a candidate coating for all described experiments, as it is the most abundant matrix in the tissues studied. Standard sulfo-SANPAH crosslinking chemistry[30] produced a monomeric collagen coating on the µTAM surface, and did not significantly affect the free expansion ratio in standard culture conditions (Supplementary Fig. 2). We did observe a small but non-significant increase in expansion variability in the 100% FBS condition, likely arising from collagen interactions with supraphysiologically high concentrations of albumin present in FBS[33]. Since in vivo interstitial albumin levels are an order of magnitude lower than in this extreme case[34], this mechanism is unlikely to impact swelling behavior in tissues. Together, these results confirm the suitability of PNiPAAM for repeated expansion cycles in situ.

### Design and characterization principles for µTAM hydrogel formulations.
To select the appropriate hydrogel formulations and model deformation, we required a conceptual framework with which to design µTAMs for tissues of different rigidity ranges. Theoretically, a complete molecular simulation from first principles could determine the stored energy density of various hydrogel formulations, but such approaches would require a combination of multiscale structural, thermodynamic, and molecular-interaction simulations with supporting characteristic measurements. As a first approximation, we instead reasoned that the dimensional expansion of compacted µTAMs is a balance between mechanical energy stored in the compressed sensors, and the mechanical work required to deform the surrounding material during expansion. Compacted µTAMs can hence be conceptualized as springs that are pre-loaded by thermodynamic expulsion of water prior to embedding in the tissue. Reducing the temperature releases this pre-strain, and the springs return to a new equilibrium position that is influenced by the rigidity of the surrounding material (Fig. 2a).

To develop finite element computational models, we approximated the stored energy density as proportional to microgel rigidity and the degree of initial compressive pre-strain. This approximation does treat any non-linear stiffening effects as a single lumped parameter, but should still provide insight into design criteria for desirable PNiPAAM properties. Simulated

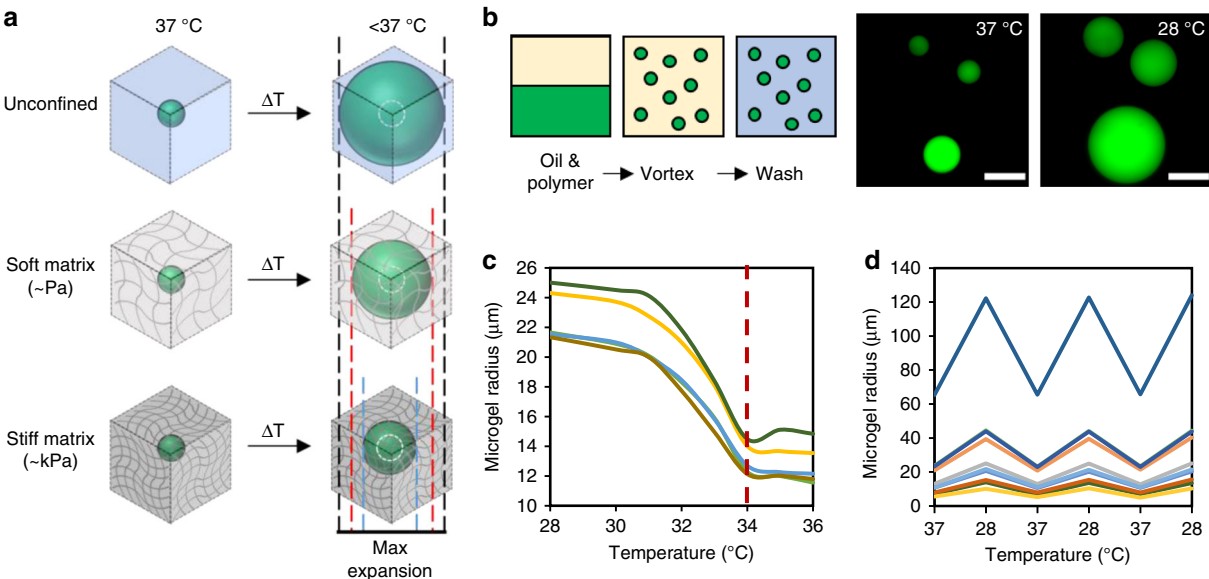

**Fig. 1 Conceptual overview of proposed technique to measure local residual elasticity. a** Poly N-isopropylacrylamide (PNiPAAM) hydrogel droplets reversibly expand and collapse based on temperature. PNiPAAM microgels can be compacted at tissue culture temperatures of 37 °C and embedded in tissues of interest, where they will keep their contracted state while the tissue is maintained in culture conditions. Reducing the temperature below the lowest critical solution temperature triggers the microgels to expand. The degree of the expansion permitted depends on the rigidity of the surrounding porous material. The expansion ratio of the sensor can hence be used to determine highly localized measurements of internal tissue residual elasticity after creep, at or near tissue culture conditions. **b** To fabricate the hydrogels, an oil/water vortex emulsion technique is used to produce polydisperse spherical microscale temperature-actuated mechanosensors (μTAMS). **c** Swelling transitions between expanded and compacted states occur at 34 °C, which can be **d** reproducibly observed over multiple temperature cycles. Different colors represent individual microgels in (**c**, **d**). Scale bar = 50 μm. Representative images are consistent over three batches of μTAMs.

spherical μTAMs of defined stiffness were isotropically pre-strained and placed within an encapsulating linear elastic material. When the pre-strain is released, a characteristic negative sigmoidal curve for μTAM expansion is produced as a function of encapsulating tissue stiffness (Fig. 2b). This is reasonable, as μTAM expansion should asymptotically approach the free expansion ratio in sufficiently soft tissues, and the completely compressed size in excessively stiff tissues. Increasing the mechanical rigidity of the μTAMs while maintaining the pre-strain levels increases the stored strain energy, shifting the sigmoidal measurement curves to provide greater sensitivities for stiffer tissues. Similar results were observed when increasing the pre-strain while maintaining μTAM mechanical rigidity. Hence, tuning the μTAM expansion ratio and mechanical rigidity can together be theoretically used to optimize stored mechanical energy in the sensors, to make measurements with desired sensitivities to tissue stiffness.

**Sensor calibration and validation in engineered tissues.** To experimentally test the trends expected through simulation, we encapsulated μTAMs in stiffness-tunable polyacrylamide tissue phantoms (Supplementary Table 2). Polyacrylamide exhibits linear elastic mechanical properties over a large strain range[30], making it an ideal phantom material for these tests. Although the PNiPAAM formulations had similarly high mechanical stiffness in their compacted states, we were unable to independently tune the expanded stiffness and expansion ratio of the μTAMs (Supplementary Fig. 3), making it difficult to predictively tune the lumped strain energy parameter underlying the model. However, low- and high- polymer content formulations were tested, and demonstrated the expected negative sigmoidal curve for increasing tissue stiffness (Fig. 2c). Based on these experimental results, we selected the 3% NiPAAM/0.2% bisacrylamide μTAM formulation for all described experiments, as it displayed the highest

measurement sensitivity within the expected stiffness ranges for soft tissue. We then determined the error associated with each individual μTAM measurement by comparing the μTAM-reported residual elasticity with the known stiffness of the tissue phantom, and empirically found that measurement errors can be modeled as linearly increasing with measurement values (Fig. 2d).

To verify that the μTAMs work as expected in an engineered tissue, we embedded them in multicellular spheroids (Fig. 3a, b), which are commonly used to model three-dimensional, diffusion-limited, and high cell-density tissues[35]. A model T47D cell line suspension was mixed with μTAMs, and formed into spheroids by aggregation[36]. As a first demonstration of μTAM stiffness sensing, we measured sensor expansion before and after tissue crosslinking through paraformaldehyde fixation, which we verified would not affect μTAM operation (Supplementary Fig. 4A). Embedded μTAMs remained circular in both their compacted and expanded states, in both live (soft) and fixed (stiff) tissues, indicating that the expansion force generated is sufficiently large to overcome small gradients of residual elasticity that may exist around each sensor (Supplementary Fig. 4B). All measurements were taken after μTAMs reached their equilibrium sizes (~30 min, Supplementary Fig. 4C), and hence all measurements reported are of residual elasticity after viscous tissue creep. The average residual rigidity increased significantly after fixation, demonstrating that the sensors function as expected (Supplementary Fig. 4D, E). These results confirm that water transport, even within densely crosslinked spheroids, is sufficient to swell the μTAMs, and that the μTAMs function as expected in an externally manipulated biological model system.

**Internal spheroid mechanics differ with cell aggregation method.** Since spheroid architecture can be internally heterogeneous, we asked whether μTAMs can be used to determine

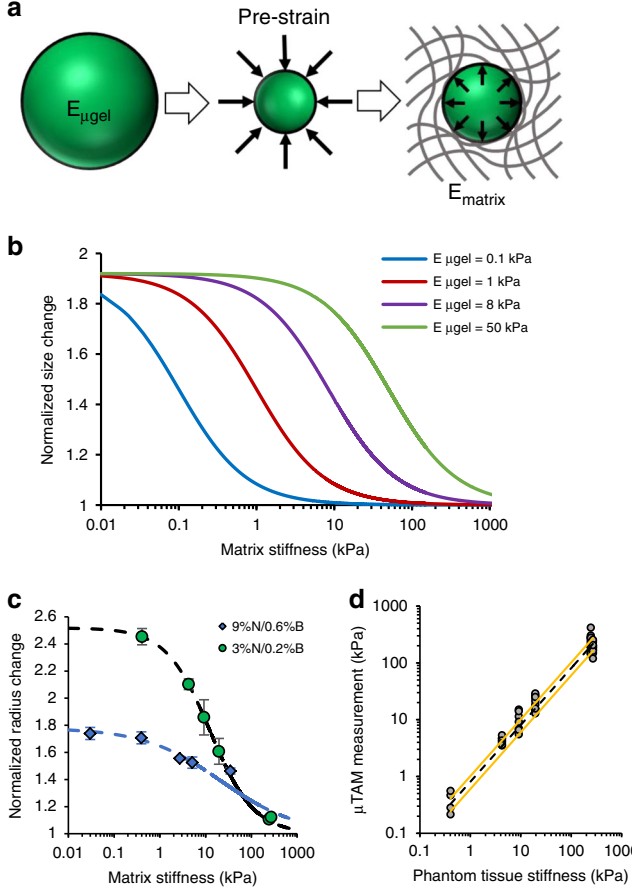

**Fig. 2 Modeling and characterization of μTAM expansion. a** μTAMs can be modeled as pre-strained springs when compacted, which then deform the surrounding matrix when the pre-strain constraint is removed. **b** Simulations using this conceptual approach indicate that μTAMs sensitivity to the stiffness of the surrounding matrix can be tuned based on stored strain energy in the μTAM, which depends on μTAM rigidity over the actuation stroke length and applied pre-strain. A characteristic sigmoidal curve is observed with maximized measurement sensitivities in distinct measurement regimes. **c** Empirical characterization data demonstrate similar sigmoidal behaviors base on μTAM polymer formulation (Supplementary Table 1; data presented as mean ± standard deviation (SD); $n = 6$–11 μTAMs). Dashed line shows simulated data from a sigmoidal data fit with iteratively optimized parameters (Eq. (2); Supplementary Table 3). **d** Multiple μTAM measurements of residual matrix elasticity are compared against rheological measurements of matrix stiffness to determine the precision of each measurement. A linear relationship between matrix stiffness (black dashed line) and measurement precision (yellow bounding lines) was observed, and used as a model to estimate the error in all subsequent measurements.

whether spheroid fabrication methods affect internal tissue mechanics. We hence formed 400–500 μm diameter spheroids containing 1–3 μTAMs from HS-5 fibroblasts (Supplementary Fig. 5A, B), using an aqueous two-phase system (ATPS) that confines cells to a small phase-separated liquid volume[37]; and a micropocket-based system in which cells passively settle into and aggregate in hydrogel cavities (Fig. 3a)[36]. These two techniques both rely on cell-driven aggregation and compaction within confined volumes, and should hence produce reasonably similar structures. No significant differences in internal cell density patterns were found in H&E-stained histology sections of the two spheroid types (Supplementary Fig. 5C–E). However, in the ATPS-formed spheroids, circumferential cell alignment increased

(Supplementary Fig. 5F–G) and was accompanied by a distinctive f-actin ring structure at the spheroid periphery (Fig. 3b). We then asked whether μTAMs might capture mechanical differences arising from these observed structural differences (Fig. 3c). Significant mechanical heterogeneity is observed across the spheroids in both cases, with measurements ranging from 0 to 13 ± 2.7 kPa in micropocket spheroids and 0 to 22 ± 4.6 kPa in ATPS spheroids (Fig. 3d), with no clear spatial patterns observable based on position within the spheroid. This broad range of residual stiffness likely reflects heterogeneity of internal architecture at these length scales within the spheroids, which is quite consistent with histology sections and with previous reports of cell heterogeneity within spheroids[30,38]. When pooled together, spheroids formed through ATPS-induced aggregation exhibited significantly higher internal residual rigidity than those formed via micropocket-based aggregation (Fig. 3e).

Hence, conceptually similar fabrication methods produce spheroids with distinct internal tissue mechanics, and while the cause of these subtle differences remain uncertain, they may arise from small osmotic compressive pressures exerted by the dextran on the spheroids in the ATPS method[39,40]. Speculatively, these differences could spatially influence cell behavior within the spheroid, which may contribute to explaining why biological findings vary considerably between research labs that use spheroids formed via slightly different methods[41]. In general, these experiments establish the utility of μTAMs in spatially characterizing internal mechanical rigidities that arise in 3D tissues.

**Internal stiffness levels of engineered tumors vary with cell type.** We next asked whether μTAMs could resolve conflicting reports regarding the stiffness of metastatic and non-metastatic cancer tumors. Invasive cancer cells themselves are well-established to be more mechanically compliant than non-invasive cell types[42], and compliant tumors are associated with local recurrence and metastasis[43,44]. However, clinical evidence suggests that metastatic likelihood increases with tissue stiffness[14,45], and external mechanical stiffness is known to promote cell migration and invasion in vitro[46–48]. Other studies suggest that the internal stiffness of invasive tumors is more heterogeneous than quiescent tumors[49]. In all cases, these observations were made using either bulk mechanical characterization, or through surface mapping of cut tissue sections, which is known to release mechanical stress[44]. Here, we aimed to use the μTAMs to characterize the internal mechanical heterogeneity of live engineered tumors generated from differently aggressive breast cancer cell lines.

Using the micropocket-formation method, we produced similarly-sized spheroids with embedded μTAMs from human metastatic breast cancer cell lines (Fig. 4a, Supplementary Fig. 6) that we have previously established to be non-invasive (T47D) and invasive (MDA-MB-231) in collagen hydrogels over 2 days in culture[36]. While the residual internal elasticity of the spheroids did vary within spheroid populations, this was not correlated with spatial position within the spheroid (Fig. 4b). The average residual elasticity was significantly greater in invasive spheroids (Fig. 4d), and reached unexpectedly high values at some sites (295 ± 62 kPa). Grubb's test confirmed that these readings were not outliers, and nearly a third of the measurements fall into a high-rigidity regime (Fig. 4c). Hence, some fraction of cells within the invasive spheroids experience extremely high local rigidities, perhaps resolving the contradictory needs for high-stiffness to prime the mechanical invasive machinery of invasive cell types, while allowing the cells to be sufficiently soft to invade through the surrounding matrix.

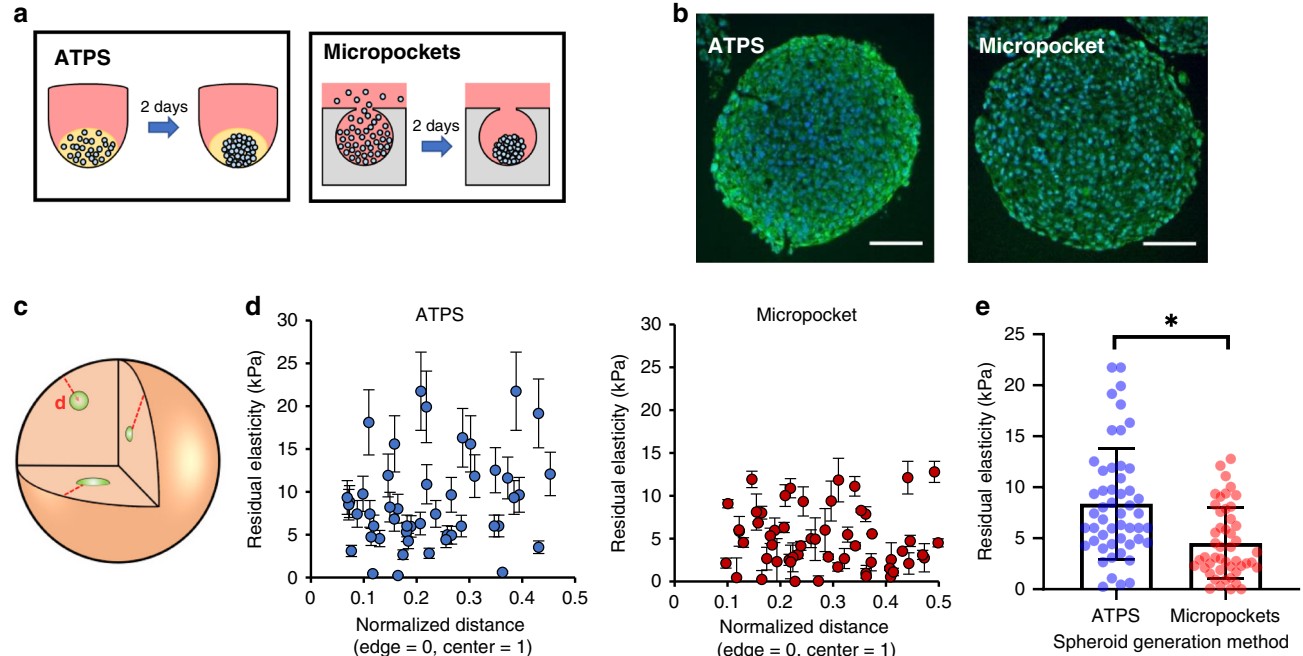

**Fig. 3 Distinct internal spheroid mechanics arise based on tissue formation technique. a** HS-5 fibroblast spheroids can be formed using a printable aqueous two-phase system (ATPS), in which cells are confined within a small droplet of immiscible liquids or by confining cells within a small cavity in a hydrogel where they passively aggregate. **b** These techniques produce grossly similar spheroids, with subtle distinctions in internal architecture as assessed by tissue sectioning and staining (green = f-actin; blue = nuclei; scale bar = 100 μm). **c** PNiPAAM microgels can be randomly incorporated into 3D multicellular spheroid cultures during the tissue formation process. **d** Pooled μTAM measurements across multiple spheroids show no obvious patterns of internal residual elasticity based on spatial location within the spheroid. Data presented as measurement ± expected error. **e** Significant differences are observed in average internal rigidity between the two formation techniques. Data presented as mean ± SD for $n = 48$ and 56 individual μTAM readings across 30 ATPS and 40 micropocket spheroids respectively for (**d**, **e**) over three independent experiments. * denotes $p = 0.0022$ for an unpaired two-tailed $t$-test. Representative spheroid images from one of the three independent HS-5 spheroid generating experiments of each method.

Although these observations of residual elasticity within spheroids is significantly higher than previously reported, these results are quite consistent with previous studies. Fresh metastatic tumor tissue sections probed by atomic force microscopy were found to be quite heterogenous, containing stiffer regions (up to 16 kPa), compared to non-invasive tumors[49]. The internal mechanical stress state of tissues can be very high when measured in live samples[50], and tissue sectioning is well-established to release these stresses and disrupt the active contractility of cells[44], which would otherwise increase rigidity in non-linear biological materials[51]. Our findings demonstrate the extent of this effect, further supporting the need for mechanical measurements without disrupting live tissue architecture. Furthermore, non-disruptive live techniques, such as quantitative ultrasound have previously demonstrated internal tumor stiffness measurements up to 150 kPa[52], albeit at considerably lower spatial resolutions. It is, therefore, likely that these measurements reflect highly focal and considerably larger rigidities that are blurred over a larger region. Hence, the measurements obtained with this technique are reasonable, and suggest that mechanical stimuli that promote invasion within tumors may be provided by small groups of cells within the spheroid population.

**Long-term measurements of internal tumor rigidity in animal models.** We then asked whether our findings extend to in vivo models. Previous studies have demonstrated that tumors can macroscopically soften, stiffen, or stay the same compared to adjacent normal tissue[43], but whether this reflects macroscale tumor organization or microscale rigidity remains unclear. Furthermore, the surrounding stromal tissue stiffens with disease

progression[46], and it is challenging to eliminate those contributions when macroscopically probing live whole-tumor mechanics. Therefore, measuring mechanics within the living tumor itself, at length scales and stroke lengths relevant to individual cancer cells may yield new insights into tumor mechanobiology.

We injected immune-competent BALB/c mice with collagen-functionalized μTAMs and a 4T1 metastatic cancer cell line that has been well-established to initially form local tumors in the mammary fat pad and spontaneously transition through the metastatic cascade with invasion to distal sites over time[53]. We confirmed that tumors grew rapidly in the mammary fat pad, degrading mammary gland tissue architecture by replacing fat tissue and lymph nodes with solid tumor, and that within 3 weeks, a heterogeneous architecture indicative of advanced disease was observed (Fig. 5c and Supplementary Fig. 7). While initially clustered along a well-defined wound track after injection, μTAMs dispersed within the tumor as the disease progressed (Fig. 5b). No signs of additional fibrosis or inflammation were observed between sham animals injected with PBS only, and those injected with PBS and μTAMs, suggesting excellent biocompatibility of the μTAMs (Supplementary Fig. 8).

Mice were sacrificed weekly, and the excised fat pads were immediately placed in a PBS bath to control tissue temperature for stiffness measurements (Fig. 5a). Only those μTAMs away from the excision wound edge were selected for analysis, to avoid measurements in regions affected by tissue stress release. These sensors were fully incorporated into the tumor tissue and retained their ability to swell and compact with temperature changes (Fig. 5d). Although the mean measurements of internal tumor rigidity did not change significantly over 21 days, we observed a significantly different distribution of measurements as the tumor

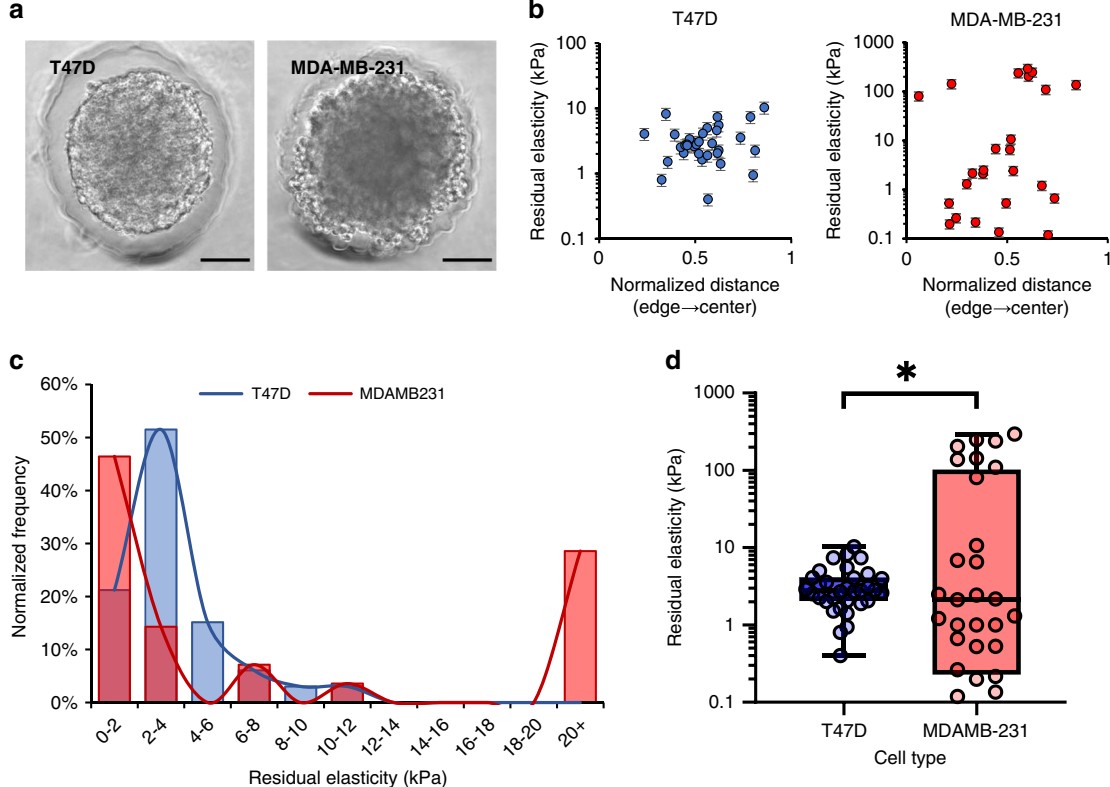

**Fig. 4 Residual elasticities within engineered tumors varies based on cell type. a** Spheroids generated from aggressively invasive MDA-MB-231 and less aggressive T47D metastatic breast cancer cell lines. Scale bar = 100 μm. **b** Spatial variation of internal residual elasticity in tumor spheroids of each cell type (data presented as measurement ± expected error, pooled from 25 to 27 spheroids with 1–2 μTAMs embedded in each). **c** Histogram of measurement data demonstrates that a significant fraction of μTAMs in the MDA-231 spheroids register high residual rigidities. **d** The average residual elasticity within spheroids are significantly different based on cell type. Box plots indicate the median and 25th to 7th percentiles, and the whiskers span the range. **b–d** *p = 0.0074 (unpaired two-tailed t-test) and n = 33 and 28 μTAM measurements for T47D and MDA-MB-231 spheroids, respectively).

progressed from day 7 to day 21 (Fig. 5e; *p = 0.022), with some sites stiffening between 25 and 50 kPa. The probability of measuring these high values in the sham control experiment are between 0.25% and 1.5%, based on descriptive z-score statistics. Certain regions within the mouse tumors were, therefore, much more rigid than the overall tumor by day 14, which matches both our findings that local mechanical heterogeneity increases in invasive engineered tumors (Fig. 4), and observations of increasing architectural heterogeneity and stromal organization as the tumor overtakes normal tissue (Fig. 5c).

While it must be recognized that injection-based models may create tumor architectures that are different from spontaneously-arising tumors, these results do demonstrate that significant differences in mechanical heterogeneity accompany tumor progression in vivo, and correlatively suggests that highly-focal sites of rigidity may be sufficient to provide a mechanical stimulus for disease progression. More broadly, in diseases such as cancer where only a few aggressive cells are required to initiate metastasis, the ability afforded by μTAMs to study highly localized mechanical microenvironments could ultimately provide an improved understanding of subpopulation-driven transitions between quiescent and malignant tumors.

## Discussion

μTAMs provide an opportunity for high spatial-resolution measurements of residual elasticity after creep in a wide range of living, three-dimensional tissues. Rather than measurements of global tumor mechanics, as has previously been reported[54,55], the

tunable size of these sensors allows interrogation of mechanical properties at multiple length scales relevant to that of a biological cell, enabling an improved understanding of the local microenvironment that cells would experience. Furthermore, whereas sensitive analysis techniques such as atomic force microscopy[56] and microrheology[57] may capture these spatial resolutions they do not approximate the stroke lengths generated by real cells, and hence measure mechanical properties of the material in a strain regime that may or may not be relevant to cellular mechanosensing and microenvironmental interrogation.

The proof-of-concept experiments developed here together demonstrate that at these cell-relevant length scales, the mechanical microenvironment in 3D tumors is far more heterogeneous than generally expected, and our findings together suggest that microscale 'hot spots' of rigidity develop as tumors progress towards an invasive, pre-metastatic phenotype. Given the well-established sensitivity of cancer cells to rigidity of the local microenvironment[46,58,59], these studies broadly demonstrate that fine spatial resolution is necessary to describe the mechanical evolution of tumors as diseases progress.

μTAMs present some limitations that require careful consideration. First, the measurements obtained with this technique cannot be quantitatively compared with those obtained from more conventional approaches, for materials that exhibit nonlinear mechanical properties, such as viscoelastic tissues[60]. Our current measurements of residual elasticity after creep can only be used to extract a stiffness modulus for materials that deform without a time-dependent component. Similarly, more advanced measurements that capture both mechanical stiffness and applied

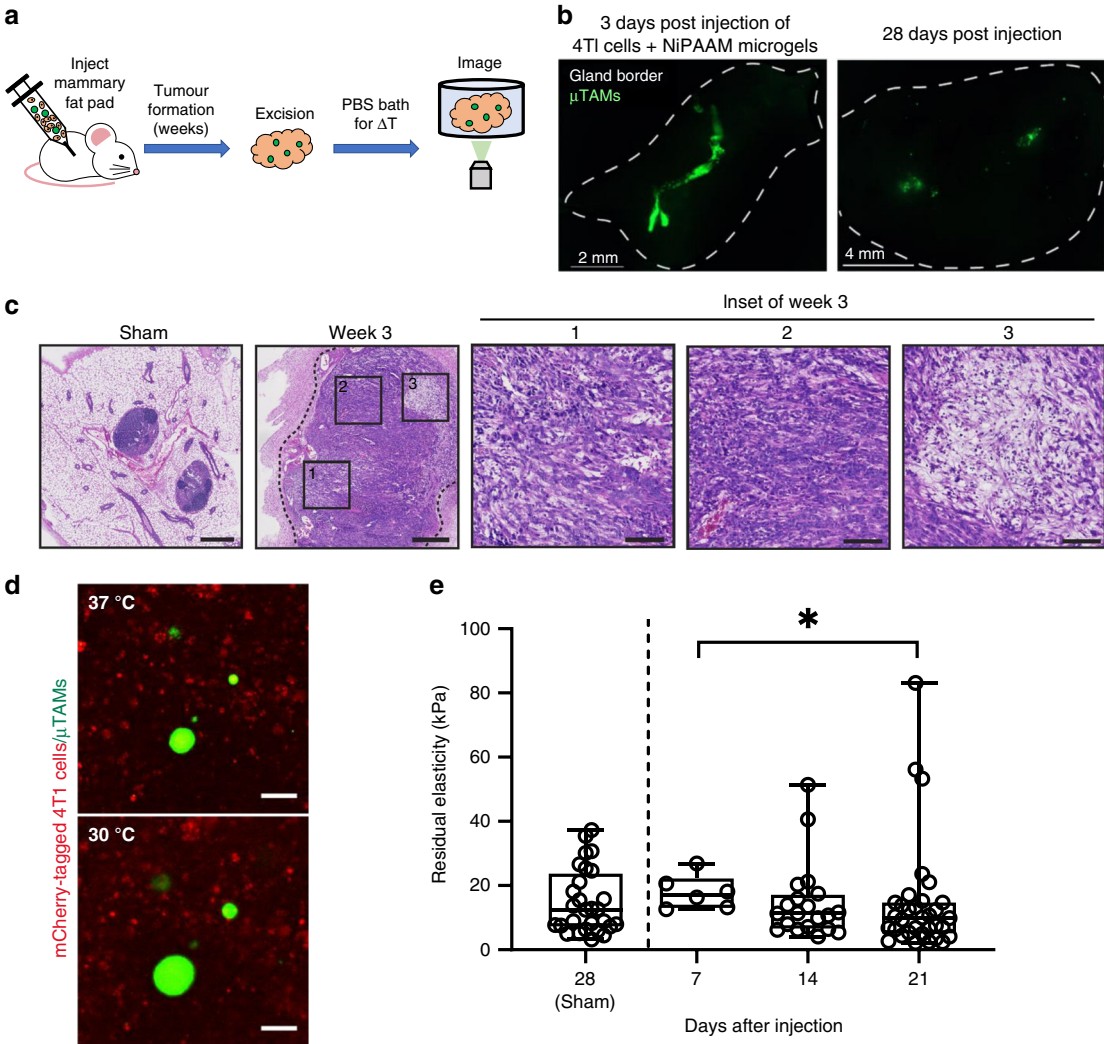

**Fig. 5 Measurements of residual internal tumor elasticity in a mouse cancer model. a** μTAMs were co-injected with mCherry-labeled T41 breast cancer cells into the mammary fat pads of female mice, and allowed to form tumors over several weeks. At various time points, tumors were excised and imaged in a temperature-controlled saline bath. **b** μTAMs are initially clustered together after injection, but disperse as the tumor develops over several weeks. **c** H&E stained tissue sections of excised fat pads at week 3 shows recovery of normal tissue architecture immediately around the needle injection site (sham, no cancer cells), and an absence of normal architecture in the tumor model. (Insets) Considerable variability in tissue cellularity is observed in distinct regions of the tumor after 3 weeks. **d** μTAMs are interspersed with mCherry-labeled 4T1 cells in the mammary fat pad, and change size when the temperature is decreased (Scale bar = 50 μm). **e** Comparison of residual elasticity within tumors indicates an increasing number of high-rigidity measurements as the cancer progresses towards metastasis, and a significant difference in measurement distributions between day 7 and day 21 of tumor progression. Box plots indicate the median and 25th to 75th percentiles, and the whiskers span the range. $n = 28, 6, 20,$ and 33 individual μTAM stiffness measurements in sham, post-injection day 7, 14, and 21, respectively. (*$p = 0.022$ by non-parametric two-tailed Mann–Whitney test to compare the distribution of ranks between groups). Representative images derived from 6 animal replicates for each time point with both left and right intraductal mammary injections to generate separate tumors.

strain, such as internal solid stress[44] also cannot be extracted. Second, μTAMs may be sensitive to confounding local factors such as pH. This is unlikely to affect the present experiments, as PNiPAAM does not behave significantly differently between pH 5–8[61], and tumors have internal pHs between 7.0 and 7.2[62], but should be considered carefully for other tissues. Third, the requisite thermal cycling could itself influence tissue stiffness. Although previous studies have demonstrated that cellular rigidity is not significantly affected between 21 °C and 37 °C[63], repetitive expansion of the sensors may theoretically induce local structural changes via damage mechanisms. To mitigate these issues in this study, we only make single measurements from each μTAM, and additional studies would be required to determine if local stress changes affect the biological systems. Fourth, the

presence of these sensors itself may affect cell behavior, as they do provide a foreign, hard surface in their compacted state. In our experiments, the hard surface presented by compacted μTAMs recapitulate microcalcification that occurs naturally in breast cancer[64]. Hence the differential responses between tissues and across timepoints in our experiments still allows us to conclude that focal stiffening is associated with invasive phenotypes. More broadly, the ability to functionalize the surface with candidate matrix molecules provide further opportunities to minimize any foreign body response.

We envision broad utility for this technology in understanding cell-scale stiffness evolution in tissues, particularly given some simple future design modifications. Developing polymer engineering strategies to tune stored strain energy through

independent manipulation of expansion ratio and sensor stiffness would enable precise manipulation of actuation force and stroke length, to better simulate mechanical interrogation by specific cell types and tune the sensors for various applications. While thermal activation was a relatively easy first step, other smart material triggers may be introduced that are faster and less disruptive, particularly for in vivo imaging. Finally, incorporating alternative imaging agents such as MRI or X-ray contrast agents would facilitate deep tissue imaging, allowing us to develop a tissue-scale cellular perspective of the local mechanical microenvironment during the highly complex processes of development and disease progression.

## Methods

Unless otherwise stated, all cell culture materials and supplies were purchased from Fisher Scientific (Ottawa, ON), and chemicals from Sigma Aldrich (Oakville, ON).

**μTAMs fabrication.** Separate solutions of 6% (w/v) polyglycerol polyrincinoleate surfactant (PGPR 4150; Palsgaard, 90415001) in kerosene; 1% (w/v) ammonium persulfate (APS) in phosphate-buffered saline; and a prepolymerized PNiPAAM solution following Supplementary Table 1 (excluding 1% APS) were each prepared in individual glass test tubes with 1–2 mL of each solution in their respective tubes. Volumes within the test tubes are fairly flexible, provided there is a matched or excess volume within the kerosene tube to create a bath. A magnetic stir bar was placed within the kerosene test tube. To purge the system of oxygen, a rubber septum stopper were used to seal each tube and nitrogen gas was bubbled through each liquid for at least 20 min using a 25G non-coring needle, with a second needle to vent the tubes to atmosphere. Microspherical gels were formed by drawing the desired amount of 1% APS solution into a syringe and dispensing it into the sealed test tube containing PNiPAAM components. The mixture was immediately vortexed and transferred into the kerosene bath with another syringe. An emulsion was made by vortexing the kerosene/PNiPAAM mixture for 5–10 s. Droplets were prevented from coalescing by gentle magnetic stirring for 20 min as the μTAMs polymerized. To facilitate washing and recovery of the μTAMs, the emulsion was aliquoted into several 1.5 mL microcentrifuge tubes. Each washing step included centrifugation at $14,800 \times g$ for 3 min, supernatant aspiration and μTAM resuspension with the appropriate solution. The μTAMs were first washed with fresh kerosene three times to remove the PGPR4150 surfactant, and then with PBS three times to recover the microgels in an aqueous phase. Finally, μTAMs were stored at 4 °C in PBS overnight to allow gels to swell to equilibrium before further use.

**μTAMs surface functionalization.** μTAMs were suspended in a 0.05 mg/mL solution of sulfoSANPAH (GBiosciences # BC38) in PBS and irradiated under 36 W UV light for 4 min. The solution was aspirated and the μTAMs were washed once with PBS before being incubated in 0.05 mg/mL solution of collagen I (Advanced Biomatrix PureCol #5005B) in PBS overnight at 4 °C. Gels were then washed with PBS and stored at 4 °C. Prior to embedding or injection into tissues, μTAMs were UV sterilized for 45 min (36 W UV source).

**Stiffness-tunable tissue phantoms.** Polyacrylamide hydrogels were fabricated on glass coverslips with embedded μTAMs to calibrate sensor measurements. Hydrogel-releasing hydrophobic glass slides were prepared by coating RainX onto 75 × 50 mm glass slides. Glass coverslips were silanized to bind polyacrylamide by immersion in a 0.4% 3-(trimethoxysilyl) propyl methacrylate (MPS) in acetone for 5 min, washed with fresh acetone for 5 min, and air dried.

To embed μTAMs into polyacrylamide tissue phantoms, polyacrylamide pre-gel solutions were made according to Supplementary Table 2 with a small volume of PBS replaced by an equal volume of μTAMs in PBS. The complete pre-gel solution with PNiPAAM microgels was pipetted onto a hydrophobic glass slide in multiple 127 μL drops to produce a 0.5 mm thick gel when a silanized 18 mm round coverslip was place on top of each drop. The solution was left to polymerize on a slide warmer set to 45 °C for 10 min. This ensures that μTAMs enter the tissue in their compacted state. After polymerization, the coverslips with the attached hydrogel were peeled off the glass slide with tweezers and placed in a multi-well plate. All hydrogels were washed three times with PBS and left to equilibrate overnight in a 37 °C incubator before thermal cycling and imaging.

**Cell culture.** Human HS-5 fibroblasts (ATCC CRL-11882); and T47D (ATCC HTB-133) and MDAMB-231 (ATCC HTB-26) breast cancer cell lines were cultured in Dulbecco's modified eagle media with 10% fetal bovine serum (FBS) and 1% anti/anti (complete media). Cells used for mice experiments were Mouse 4T1 (ATCC CRL-2539), which were cultured in RPMI 1640 (Wisent) with 10% FBS, 1% sodium bicarbonate, 0.5% sodium pyruvate, and 0.5% HEPES. When the cells reached at least 80% confluence (70% for 4T1 cells to maintain tumorigenic characteristics), they were detached using 0.25% trypsin-EDTA and either subcultured into a new culture vessel at a 1:10 ratio or used as a single cell suspension for experiments.

**Spheroid formation via aqueous two-phase systems.** Spheroids formed via aqueous two-phase systems (ATPS) were grown in a non-adhesive 96-well round bottom plate following previously published techniques using a robotic liquid handler (Gilson PipetMax, Mandel, Guelph ON)[37,65]. Briefly, a 0.2% (w/v) solution of Pluronics F108 in PBS was pipetted into each well and incubated for 1 h at room temperature (23 °C). The solution was aspirated, and the wells rinsed with reverse osmosis (RO) water before air drying. Plates were sterilized under UV light for 45 min prior to use. Stock solutions of 6% (w/v) polyethylene glycol (PEG) in complete media; and diluted to 5.4% in water prior to use. A cell-laden dextran (DEX) solution was prepared by mixing 85 μL of a 15% (w/v) dextran in PBS solution with 15 μL of a $17 \times 10^6$ cells/mL suspension of HS-5 fibroblasts. To incorporate PNiPAAM microgels into the spheroids, 1–3 μL of the functionalized microgel suspension was mixed into the cell-laden dextran depending on the desired microgel to spheroid ratio. 50 μL of the PEG solution was dispensed into each well of the non-adhesive 96 well-plate, and 1 μL of cell-laden DEX was carefully dispensed slightly above the bottom of each well. The plates were carefully transferred to a cell culture incubator (5% $CO_2$, 37 °C) for 1 h before adding 75 μL of complete media and growing the spheroids for two days.

**Spheroid formation via micropocket hydrogel cavities.** Spheroids were formed in polyacrylamide micropockets using previously published protocols[36]. Poly-acrylamide micropockets were cast using the 12% acrylamide/0.24% bis-acrylamide formulation. Approximately 125 μL of the prepolymer polyacrylamide solution was dispensed over a 3D printed mold containing ~200 spherical structures of 0.5 and 1 mm diameters across the surface area of a 12 mm coverslip to generously fill the mold. An MPS-treated 18 mm coverslip was placed on top of the mold, and the hydrogel was allowed to polymerize for 10 min. The polymerization grafted the polyacrylamide hydrogel to the coverslip, which was then gently separated from the 3D printed mold, and washed three times in PBS. Gels were stored at 4 °C in PBS to equilibrate overnight, and sterilized under UV light for 45 min. PBS was aspirated prior to loading the micropocket gels with cells. A mixture containing 100 μL of a $15 \times 10^6$ cells/mL suspension of the desired cell type with 1 μL of functionalized μTAM suspension was distributed over each hydrogel. The cells were left to settle into the micropockets for 5 min before submerging the entire polyacrylamide micropocket device in complete media. Spheroids then formed over 2 days in a standard cell culture incubator (5% $CO_2$, 37 °C).

**Mouse breast cancer model.** Mice were housed at the Goodman Cancer Research Center animal facility in adequate enclosures as described in the Canadian Council on Animal Care guidelines for mice. Specifically, the rooms go through a 12 h light/dark cycle from 7 am to 7 pm and ambient temperatures were set to 22 °C with humidity kept at 40%. Up to 5 adult mice are kept in each cage dressed with a bedding of corn and Enviro-dri (Cedarlane; Burlington, ON). Cages are supplied with enough food for 2 weeks with weekly top ups and water is freely available through water bottles fitted with animal drinking valves in the cages.

All procedures were performed in accordance with the animal care guidelines by the Canadian Council on Animal Care after obtaining ethics approval from the Animal Resource Centre of McGill University. For each replicate, a set of 4 female BALB/c mice (Charles River) at 8–10 weeks of age were randomly allocated a condition (sham, week 1, week 2 or week 3). Mice were anesthetized under isoflurane gas, as the 4th and the 9th mammary fat pads were injected using a 22G needle (Becton Dickinson) attached to a Hamilton syringe. Each gland was injected with a suspension of mCherry-labeled 4T1 cells at $5 \times 10^5$ cells/mL with different concentrations of μTAMs in 25 μl of sterile PBS. The 4T1 tagged cells were generated using lentivirus and the lentivector pWPI-mCherry. Sham condition mice were injected only with a suspension of μTAMs in 25 μl of sterile PBS and left for 3 weeks

Mice were euthanized by cervical dislocation under isofluorane anesthesia. At the indicated time points, injected mammary fat pads or tumors were surgically isolated and immediately rinsed in sterile PBS. Tumors exceeding 5 mm in thickness were sectioned to layers 4 ± 1 mm in thickness to facilitate bead visualization and rinsed 10 times in sterile PBS. Tissue was immersed in sterile PBS in a 2-well chambered cover glass (Nunc™ Lab-Tek™) for immediate imaging.

**Temperature-controlled imaging and μTAMs size analysis.** Polyacrylamide phantoms, and multicellular spheroids were mounted in a Chamlide imaging chamber and submerged with 300 μL of PBS before being placed on a controlled stage warmer (Ibidi). Images were taken on an Olympus IX-73 microscope under epifluorescence (Olympus, X-CITE 120 LED), with an sCMOS Flash 4.0 Camera and Metamorph software (version 7.8.13.0), and automated software (Zaber) to record and return to specified positions. Samples were mounted in a live-cell imaging chamber (Ibidi), and imaged initially at 37 °C and during cool-down to room temperature at 30 min intervals to ensure temperature equilibration and complete sensor size change. Live mouse tissue explants were imaged with an LSM700 laser scanning confocal microscope with a 20 × 0.8NA objective lens and ZEN software (Zeiss) in a temperature-controlled environmental chamber. The tissue was then

incubated at 37° C for 1 h, and the same positions were re-imaged using the same imaging parameters. Images were deconvolved using the iterative deconvolve 3D plugin[66] and a point spread function generated by imaging 0.19 μm green TFM beads in identical imaging conditions as the μTAMs.

μTAMs that were damaged (missing chunk or fragment), in contact with an adjacent sensor, or partially exposed outside of tissue were excluded from measurements. μTAMs that were less than 10 μm in diameter were excluded from stiffness analysis to reduce measurement error. All μTAM images were analyzed in FIJI by manually drawing a fitted ellipse around the μTAM and measuring the Feret's diameter for the microgel size, as well as shape descriptors for the circularity of the μTAM which was calculated within the software as:

$$\text{Circularity} = \frac{4\pi \times \text{Area}}{\sqrt{\text{Perimeter}}} \quad (1)$$

Conversion between μTAM size change ratio was done using the modeling curve and the iterated best fit values for parameters specified in Supplementary Table 3:

$$\frac{D_{30}}{D_{37}} = \alpha - \frac{\alpha - 1}{1 + \left(\frac{E_{\text{matrix}}}{E_{\text{bead}}}\right)^{-\beta}} \quad (2)$$

where $D_{30}/D_{37}$ is the observed expansion ratio between the μTAM diameter at 30 °C (expanded) over its diameter at 37 °C (compacted), $\alpha$ is the expansion ratio of μTAMs in free solution, $E_{\text{matrix}}$ is the apparent stiffness of the matrix, $E_{\text{bead}}$ is the apparent stiffness of the μTAM, and β is a lumped parameter estimated by curve fitting that captures friction, surface penetration, and other losses.

**Shear rheometry for bulk characterization of hydrogels**. The stiffness of each polyacrylamide and PNiPAAM gel formulation was measured using a parallel plate shear rheometer (Anton-Paar, MCR 302) in strain-controlled mode. Hydrogels fabricated for shear rheology were made by sandwiching 113 μL of the complete pre-gel solution (compositions provided in Supplementary Table 2) between two 12 mm MPS-treated coverslips. After 10 min, the sandwiched polymerized hydrogels were placed in a multi-well plate and submerged in PBS. After three washes, the gels were left to swell overnight at 4 °C. During testing, excess PBS was dried off the top and bottom of the samples, and adhesively mounted between rheometer plates. Storage and loss moduli were recorded over a strain sweep that was run from 1 to 50% at 10 Hz and verified to be plateau within this range. The moduli values were reported as an average of all the readings. Young's modulus ($E$) was calculated using $E = 2\,G(1 + \nu)$ where $G$ is the average storage modulus, and $\nu$ is the Poisson's ratio of the hydrogel which was assumed to be 0.5 based on literature[67].

**Histology and staining**. Spheroids were fixed in 4% paraformaldehyde solution for at least 24 h at 4 °C. Spheroids in the micropockets were extracted and transferred using a clipped P1000 pipette tip placed directly over the chamber opening. Spheroids formed by ATPS were pipetted directly with a clipped pipette tip. Spheroids were collected into a 1.5 mL microcentrifuge tube with PBS embedded in paraffin blocks. Tissue blocks were sectioned at 4 μm and mounted on charged glass slides for histology.

For staining, the tissue sections were deparaffinized in xylene for 15 min and rehydrated using a decreasing ethanol gradient at 100%, 90% and 80% for 2-min intervals. Slides were washed twice with PBS for 5 min, and permeabilized in 0.1% Triton-X solution for 5 min, before two additional PBS washes. Tissue sections were blocked with 1% BSA in PBS for 30 min at room temperature (23 °C). An actin cytoskeletal and nuclear staining mixture of FITC-conjugated phalloidin (1 μg/mL) and Hoechst 33258 (1 μg/mL) in 1% BSA was applied for 20 min. The slides were washed twice in PBS and once with water before coverslip mounting using Fluoromount Aqueous Mounting Media and sealing with clear nail polish.

**Histological section image analysis**. All image analyses were performed using FIJI[68]. The cross-sectional area of the circular spheroids were segmented into 5 annuli of equal area. Cell density in each anulus was quantified with an automated nuclear count by thresholding the image to isolate the nuclei and performing a particle analysis count with a minimum of particle size of 20 μm². Nuclear orientation was analyzed by determining the difference between the expected angle for a circumferentially aligned nucleus ($\Theta_{\text{expt}}$) and the angle of the nucleus itself as determine by the particle analysis on FIJI. To get $\Theta_{\text{expt}}$, the angle at the center of a circle ($\Theta_r$) was calculated by taking the tangent angle between the $X$ and $Y$ distance of the nucleus to the center of the spheroid. $\Theta_{\text{expt}}$ was calculated by assuming spherical symmetry in the spheroid and taking the absolute value of $\Theta_r + 90°$ if $\Theta_r > 0°$ or $\Theta_r - 90°$ if $\Theta_r < 0°$.

**Statistical analysis**. $Z$-scores were used to assess the probability of obtaining measurements compared to a control population. For non-normal distributions, log transformations were used to first obtain normal distributions, which were confirmed via Shapiro–Wilks tests. Comparative data analyses of populations were performed without pre-specifying a required effect size. Datasets that were normally distributed, with similar variances between compared groups were analyzed

using unpaired $t$-tests, one-way, or two-way ANOVA to test for significance, which was set at α = 0.05. Post-hoc pairwise comparisons were conducted using the Bonferroni method. Datasets that were not normally distributed were analyzed using the nonparametric Mann–Whitney test to compare the distribution of ranks between two groups, with significance values set at α = 0.05. All statistical analyses were performed using GraphPad Prism v8.0.2 (San Diego, CA).

**Finite element modeling of μTAM expansion**. Simulations were performed using the open-source software package FEBio[69] with the pre-strain plugin[70] to apply compressive loads to simulated μTAMs prior to release within an encapsulating matrix of defined stiffness. 3D spherical geometries were used to simulate the μTAMs (unit radius) embedded in a 10× larger encompassing sphere, to simulate an infinitely large matrix. The model was meshed with hexahedral elements, and a mesh size sensitivity analysis was performed. Less than 1% variation was observed in deformation for a mesh element size of 0.16 at the μTAM/matrix interface, for an r-ratio of 1.57. Fixed displacement boundary conditions were applied to the outer matrix surface, and a tied contact interface was defined at the μTAM/matrix interface. Linear elastic material properties and initial pre-strain of the μTAMs were defined and modulated based on experimental data. Analyses were conducted using a dynamic large deformation structural mechanical analysis, and data is reported as a fold change in μTAM size for matrices of various mechanical stiffness.

**Reporting summary**. Further information on research design is available in the Nature Research Reporting Summary linked to this article.

## Data availability

Select μTAM characterization data and all μTAM stiffness measurements from spheroid and animal experiments are provided in Supplementary Tables. All additional data presented in this paper are available upon reasonable request from the corresponding author.

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

## Acknowledgements

We thank Professors Richard Leask and Milan Maric (McGill University) for rheometer access and expertise; Dr. Clement Ma (Harvard University) for consultation on statistical methods; and the Goodman Cancer Research Centre Histology Facility for assistance with processing, embedding and section tissue samples. This work was supported by the Canadian Cancer Society (Grants # 704422 and 706002) and the Canadian Institutes for Health Research (Grant # 01871-000) to C.M. and L.M., and the NSERC Discovery RGPIN-2015-05512 (C.M.). We gratefully acknowledge support from NSERC Canada Graduate Scholarship-Doctoral to S.M., Postgraduate Scholarships-Doctoral to W.L., and the Canada Research Chairs in Advanced Cellular Microenvironments to C.M.

## Author contributions

S.M. and C.M. formulated the idea behind the study. S.M., L.M., and C.M. designed experiments. S.M. and K.M conducted µTAM empirical characterization. S.M. performed experiments in spheroid cultures, and conducted stiffness analysis for all experiments. C.L. performed finite element simulations. S.A. performed µTAM injections into mice and imaging of ex vivo tumors. S.M. and W.L. performed mechanical characterization and data analysis/fitting. L.M. and C.M. provided reagents, materials, animals, and analysis expertise. S.M. and C.M. drafted the paper. All authors edited the paper.

## Competing interests

The authors declare no competing interests.
