## [Peer Review File · Nature Communications]

Reviewers' Comments:

Reviewer #1:

Remarks to the Author:

This work described a new method of using temperature-controlled phase transition of polyNIPAM microgel to measure spheroid or tissue modulus. Tissue mechanics are complex but important in many aspects of biology such as cancer and regeneration. The approach used here to testing the internal stiffness of spheroid or tissue is innovative. This manuscript also demonstrated the utility of this method in a few different models using biomaterials, spheroids, and in vivo tumors. Overall, this method is quite promising, but its limited accuracy and other limitations indicate that it is more of a semi-quantitative method for tissue mechanical measurement at the current stage.

Specific comments:

1. As the authors mentioned in the Discussion, the uTAM method is unable to resolve time-dependent mechanical properties such as viscoelasticity. Living tissues, cells and ECM are viscoelastic and exhibit different viscoelastic behaviors. The apparent elastic modulus/stiffness of viscoelastic materials is loading rate dependent, so the uTAM method cannot measure the initial elastic modulus and it mainly captures the residual elasticity. So technically speaking, the uTAM method does not measure true elastic modulus. In addition, the uTAM method is unable to accurately compare the elastic modulus/stiffness of tissues with different viscoelastic properties.
2. The spatial distribution of uTAMs seems to be highly affected by the formation process of spheroids, which can artificially skew the apparent modulus. In Fig. 3E, there are only 3 uTAMs between ~ 0.3 - 0.5 of edge to center normalized distance in the APTS spheroids, but there are > 15 data points between ~ 0.1 - 0.3 edge to center distance. On the other hand, the distribution of uTAMs in PAA spheroids are almost the opposite. Due to the large variation/error bar for each data point, this can skew the actual patterns of stiffness peaks.
3. The authors reported modulus as high as 300 kPa in invasive spheroids (MDA-MB-231). This is much higher than what has been reported before using different testing methods, especially considering the short time of spheroid formation in this study. Based on the principle of how the uTAM method works, it is expected that the stiffer the sample the larger the error (the very small size change of uTAMs in stiff tissues will generate large measurement errors). This is shown in Fig. 2D. Although there are limitations of other methods, it is not convincing to simply accept the uTAMs result of spheroids without any validation.
4. It was claimed that the sensitivity of uTAMs could be tuned (Fig. 2B simulation). However, the coupled expansion ratio and expanded stiffness of uTAMs actually makes their sensitivity not so tunable in practice. For example, in Fig. 2C, the 9%N/0.6%B condition doesn't seem to be better suited for testing stiffer samples, which contradicts the idealized simulation.
5. In the study of animal tumor models, the result was interpreted as to suggest a link between highly localized mechanical properties within the tumor and disease progression towards metastasis (e.g. line 337-348). However, the development of those inoculated tumor models is very different from that of spontaneous tumors. Such claim can be misleading.
6. The ex vivo uTAM result shows that the average apparent moduli of breast tumor tissues decrease over time and are smaller than that of sham (Fig. 5E), which is inconsistent with previous reports using different mechanical testing methods (e.g. <https://doi.org/10.1038/s41551-016-0004>; <https://doi.org/10.1371/journal.pone.0193801>).

Reviewer #2:

Remarks to the Author:

Summary:

Mok et al have developed a method to assess tissue stiffness with spatial resolution close to the size of single cells. This manuscript presents the development and validation of thermoresponsive microspheres that expand when cooled from 37°C to 28°C. The group shows that the differential expansion of the microgels can be used to determine the local stiffness of the surrounding material. This technique is then used to determine the cell-scale stiffness of cell spheroids and mammary tumors in a mouse model. There is a need for new techniques to measure local mechanical properties, and this approach could be broadly utilized by researchers in the mechanobiology field. However, the validation of the measurements made in this manuscript is not sufficient for publication and there are concerns about the robustness and reliability of the biological findings presented here.

Major Comments:

1) To demonstrate reliable measurements with this technique, the authors need to validate that uTAMs can be used for measuring elastic moduli over the range encountered in their biological assays. The authors report stiffness values up to 250 kPa in later experiments, while they have only validated the measurement range against rheometry up to 20 kPa, with a fair bit of variation even at that stiffness. Further, the authors should corroborate their local stiffness measurements using a conventional technique (microrheology, AFM, etc.) to provide more confidence in these measurements.

2) There is a huge degree of variation in the reported stiffness values for measurements in cell spheroids. For example, in figure 4B, it appears that there is a difference in stiffness of three orders of magnitude over a few micrometer length. Since these drastic spatial differences in stiffness have not previously been observed, and since these stiffness values are much higher than those reported with other techniques, the authors should show using a standard measurement technique that these measurements are accurate.

3) Many of the biological insights in this manuscript rely on inference from outlying data points. For example, in the abstract the authors state "we determine that small sites of unexpectedly high stiffness (>250 kPa) develop in invasive breast cancer spheroids, and in in vivo mouse model tumors." It appears that the evidence for this statement regarding the mouse tumors is based on two measurement locations at day 14 and three at day 21 out of what seems to be dozens of data points. There does not appear to be any significant difference in the average stiffness along the time course of the experiment.

4) The reported values for the stiffness of spheroids here differs greatly from prior reports. For example, Han et al (Nature Physics, 2019 doi: 10.1038/s41567-019-0680-8) showed that mammary spheroids were in the range of 25-100 Pa by microrheology and the Physical Sciences Oncology Centers Network (Sci. Rep. 3:1449, 2013) found the mammary cells had elastic moduli under 2 kPa by AFM indentation. Can the authors discuss the reasons for these discrepancies if they exist after more rigorous validation?

5) Spheroids and soft tissues generally exhibit a significant degree of viscoelasticity. What is the timescale over which the spheroids/tissues are cooled? It seems probable that some of the stress induced by expansion of the microgel is dissipated by the viscous components of the tissue before measurements are made. Can the authors discuss how this might affect their measurements or potential avenues to overcome this issue?

Minor Comments

- 1) Can the authors explain why there is no observed increase in tumor stiffness along their experimental time course or why the sham injection is equally stiff as the tumors? Tumors that are stiffer than mammary fat pads and that stiffen over time have been observed by many others using multiple measurement modalities.
- 2) It is not clear what the different colors represent in Fig. 1 C and D. Are these different individual microgels?
- 3) It is not entirely clear what conclusions should be drawn from the two different spheroid generation strategies in Fig. 3.
- 4) The vast majority of the data points in Fig. 4B are indistinguishable from 0. Can the authors display this data differently, perhaps with a log scale y-axis? Similarly for Fig. 4D.
- 5) The y-axis label in Fig. 4C should be changed to frequency or something similar.
- 6) Why are the fluorescent regions in Fig. 5B not spherical? How does this affect the measurements?
- 7) The y axis should again be adjusted in Fig. S3 so that the data can be interpreted.

We thank the reviewers for their time, effort, and insight in examining our manuscript. We agree with the issues raised, and have conducted several additional experiments, refined our descriptive text and presentation, and restructured our critical discussion of this work. The result, we feel, is a substantially more rigorous and insightful manuscript. Our new findings and responses to the critiques are outlined in the following point-by-point response (where appropriate, manuscript changes have been reproduced in blue text). We would like to thank the reviewers and editors for what has been an extremely rewarding peer review process, and hope that these revisions satisfactorily address the concerns raised.

Reviewer #1:

Overall, this method is quite promising, but its limited accuracy and other limitations indicate that it is more of a semi-quantitative method for tissue mechanical measurement at the current stage.

We do appreciate this overall comment and agree that the relatively large error bars at the upper end of the measurement range are visually disconcerting. However, we believe that this visual perception is driven by our decision to present all our data on linear scales. In contrast, presenting this data on a logarithmic scale (as is the standard approach when making measurements across 4 orders of magnitude), makes the relatively “large” error bars visually small. As an example of this, we present here a comparison between two identical datasets (previously Figure 4B, panel 2); on linear (panel A) and log scales (panel B):

Figure (not included in manuscript). Spatial stiffness from MDA-MB-231 spheroids data graphed on a (A) linear versus (B) log y-axis, demonstrates differences in visual perception for the same data set.

Hence, we believe that our initial choice to present data on a linear axis played a considerable role in this negative perception of limited precision (we believe the reviewer means ‘precision’ rather than ‘accuracy’). In our view, since we have (1) quantitatively determined the experimental precision of our measurement technique, (2) calibrated the measurement system

against quantitatively validated samples, and (3) included these measurement errors in our analyses, interpretation, and discussion; we believe that this therefore does allow us to claim that this is a quantitative metric.

While we had originally intended to use linear scales to clearly outline the limitations of our technique, we also appreciate that this gives the wrong impression of utility and impact.

To more clearly make our case for this tool as a quantifiable measurement system we have:

- 1) Specified the +/- measurement error clearly when referring to these values in the text
- 2) For measurements that span multiple orders of magnitude, we, have replaced linear scales with log scales (Manuscript figures 2D, 4B, 4D)

We hope that this clarifies and addresses this overall perception; and have addressed the other limitations mentioned in response to subsequent comments.

1a. As the authors mentioned in the Discussion, the μ TAM method is unable to resolve time-dependent mechanical properties such as viscoelasticity. Living tissues, cells and ECM are viscoelastic and exhibit different viscoelastic behaviors. The apparent elastic modulus/stiffness of viscoelastic materials is loading rate dependent, so the μ TAM method cannot measure the initial elastic modulus and it mainly captures the residual elasticity. So technically speaking, the μ TAM method does not measure true elastic modulus.

We agree with this excellent point, and acknowledge our previous poor choice of language. As the reviewer correctly points out, we are indeed measuring the residual elasticity of the material after creep, rather than storage or material moduli. We have therefore clarified this point throughout the manuscript (title, abstract, figure captions, axis labels, main manuscript text, SI). Furthermore, we experimentally confirmed that we are indeed measuring residual elasticity after creep deformation is completed, via a time course expansion experiment to track μ TAM size within spheroids. Our results confirm that sensor expansion stabilizes within 30 minutes, the time at which all reported measurements were taken. Hence, we are confident that the values reported in this manuscript do capture the ‘after creep’ residual elasticity.

Figure S4 C (new). μ TAM size change over time within T47D spheroids during cooling on our microscope stage. Temperature and size measurements were recorded every 5 minutes for an hour to ensure that 30 minutes was adequate time for μ TAM expansion to equilibrate, thereby ensuring residual elasticity around the μ TAM is measured.

This new data has been added to the manuscript as Supplemental Figure S4C, and referred to in the manuscript as follows:

Results > Sensor calibration and validation in engineered tissues: “...All measurements were taken after μ TAMs reached their equilibrium sizes (~30 minutes, Supplementary Fig. S4C), and hence all measurements reported are of residual elasticity after viscous creep of the tissue.”

1b. In addition, the μ TAM method is unable to accurately compare the elastic modulus/stiffness of tissues with different viscoelastic properties.

We agree, and have now ensured that we

- (1) clearly state that we measure residual elasticity throughout the manuscript
- (2) have avoided drawing unqualified comparisons with elastic modulus / stiffness; and
- (3) discussed this important limitation in the discussion section of the manuscript:

Discussion: “ μ TAMs present some limitations that require careful consideration. First, the measurements obtained with this technique cannot be quantitatively compared with those obtained from more conventional approaches in materials that exhibit non-linear mechanical properties, such as viscoelastic tissues⁶¹. Our current measurements of residual elasticity after creep can only be used to extract a stiffness modulus for materials that deform without a time-dependent component.”

2. The spatial distribution of μ TAMs seems to be highly affected by the formation process of spheroids, which can artificially skew the apparent modulus. In Fig. 3E, there are only 3 μ TAMs between ~ 0.3-0.5 of edge to center normalized distance in the ATPS spheroids, but there are > 15 data points between ~ 0.1-0.3 edge to center distance. On the other hand, the distribution of μ TAMs in PAA spheroids are almost the opposite. Due to the large variation/error bar for each data point, this can skew the actual patterns of stiffness peaks.

We are extremely grateful for this comment. While attempting to address it, we conducted multiple additional experiments and pooled these results to obtain a considerably larger dataset than before. Using this new dataset, we discovered that the trends that had seemed clear were not substantiated. We therefore retract our claims for spatial variations in stiffness, and are extremely grateful to the reviewer for helping us avoid a considerable error. The updated and expanded dataset is presented here (and in Figure 3D of the manuscript). Interestingly, the expanded dataset now provides sufficient statistical power to identify statistically significant differences in internal residual elasticity between spheroids produced using the two fabrication techniques. This new analysis is now added in as Figure 3E in the manuscript.

Figure 3D (revised). Spatial stiffness in HS-5 spheroids generated via (A) ATPS and (B) polyacrylamide (PAA) micropockets. $n = 48$ and 56 individual μ TAM readings across 30 ATPS and 40 PAA spheroids respectively.

Figure 3E (new). Significant differences are observed in average internal elasticity between the two formation techniques. * denotes $p = 0.0022$ for $n = 48$ and 56 individual μ TAM readings across 30 ATPS and 40 micropocket spheroids respectively in an unpaired t-test.

This new data supports our original conclusions that the spheroid formation method significantly influences internal tissue stiffness (magnitudes rather than spatial patterns), and is also consistent with our supporting histology sections showing more ordered tissue structures in spheroids with higher internal stiffness.

To incorporate this new data, analysis, and discussion we have:

- 1) Updated our original dataset with our expanded data set (Fig. 3D).
- 2) Added figure panel (Fig. 3E) in the main manuscript comparing the measurements of residual elasticity magnitude
- 3) Revised our abstract to reflect these points.

Abstract: "...Using the microfabricated sensors, we demonstrate that mapping internal stiffness profiles of live multicellular spheroids can be done at high resolutions to reveal broad ranges of stiffness present within the tissue, which vary with subtle differences in spheroid aggregation method."

- 4) Revised our Results > "Internal spheroid mechanics differ with cell aggregation method" section title and text to remove claims of spatial mapping of stiffness and describe the statistically significant changes observed, as follows:

Results > Internal spheroid mechanics differ with cell aggregation method: "No significant differences in internal cell density patterns were found in H&E-stained histology sections of the two spheroid types (Supplementary Fig. S5C-E). However, in the ATPS-formed spheroids, circumferential cell alignment increased (Supplementary Fig. S5F-G) and was accompanied by a distinctive f-actin ring structure at the

spheroid periphery (Fig. 3B). Since structural differences likely affect internal mechanics, we then asked whether μ TAMs might capture these differences by embedding them within the formed spheroids (Fig. 3C). Significant mechanical heterogeneity is observed across the spheroids in both cases, with measurements ranging from 0 to 13 ± 2.7 kPa in micropocket spheroids and 0 to 22 ± 4.6 kPa in ATPS spheroids (Fig. 3D). Although there are no clear spatial patterns observable based on position within the spheroid, this broad range of residual stiffness likely reflects heterogeneity of internal architecture at these length scales within the spheroids, which is quite consistent with histology sections and with previous reports of cell heterogeneity within spheroids^{1,2}. When pooled together, spheroids formed through ATPS-induced aggregation exhibited significantly higher internal residual elasticity than those formed via micropocket-based aggregation (Fig. 3E)."

3a. The authors reported modulus as high as 300 kPa in invasive spheroids (MDA-MB-231). This is much higher than what has been reported before using different testing methods, especially considering the short time of spheroid formation in this study.

We confirm that these points are correct, and have addressed them in response **3c**.

3b. Based on the principle of how the μ TAM method works, it is expected that the stiffer the sample the larger the error (the very small size change of μ TAMs in stiff tissues will generate large measurement errors). This is shown in Fig. 2D.

We would like to draw the Reviewer's attention to a significant improvement made to the manuscript: we have now extended the experimental calibration and error analysis range (Fig. 2C, D) to include the upper ranges of residual elasticity. This allows us to interpolate both accuracy and precision across our experimental datasets (for additional details, see Reviewer #2, Comment #1). Although these new experiments did not change our results, they do indicate that we can now confidently report the quantitative errors present, even for stiffer samples.

3c. Although there are limitations of other methods, it is not convincing to simply accept the μ TAMs result of spheroids without any validation.

We would first like to clarify that we have in fact validated the sensors themselves through repeated experiments in defined phantoms (Fig. 2C), and have also validated their operation in spheroids of different cell types (Fig. S4, Fig. 3B). These experiments produced measurements consistent with expectations of increased tissue stiffness through fixation (Fig. S4; T47D cells), and with tissue architecture corresponding to histology (Fig. 3B; HS-5 cells). We therefore believe that our measurement system has been validated, and is now being applied to make measurements that were not previously possible using existing techniques. To more precisely state our validated measurement results throughout the manuscript, we have made sure to add measurement errors (295 ± 62 kPa in this case) into all textual references for these values (abstract, results, discussion). This then indicates that the true value could lie within this range.

We do however also agree with and appreciate the underlying spirit of this comment, which is to support these surprising findings in spheroids with other evidence. We initially hoped to find an alternative measurement system, but (as acknowledged by the reviewer), other mechanical measurements techniques have severe limitations in this context, and cannot be directly compared to our technique. Indeed, we would like to reiterate that the *unique strength* of this work is the ability to measure residual elasticity at the length scale of individual cells, within large, living, undamaged biological tissue; which simply cannot be achieved through other means. Since we cannot confirm our measurements through quantitative means, we next aimed to find conceptual, correlative and qualitative evidence that supports these findings, as follows:

- 1) Quantitative ultrasound (shear elastography) of mechanical stiffness within intact tissues has identified sites as stiff as 166 +/- 49 kPa in human breast cancers^{3,4}. The spatial resolution of ultrasound shear elastography is approximately 150-200 microns; and these results may in fact reflect a much smaller and stiffer region that is “blurred” or averaged over a larger area. Hence, ultrasound elastography measurements do support our findings that high stiffness can arise within living, intact, cell-dense material.
- 2) We note that our measurements are made without disrupting the baseline tension exerted by cells while alive, and in three-dimensional culture. Others have measured the stiffness of nuclei in single well-spread cells, and obtained apparent Young’s modulus values of over 80 kPa⁵. Clusters of cells within spheroids can be tightly packed and well-adhered, and could hence create regions that have an even higher effective stiffness.
- 3) In previous studies from our lab, we have demonstrated that phosphorylated myosin, an indicator of active cytoskeletal contractility occurs in small regions within a spheroid (Lee et al., 2019; Figure 5h).

[REDACTED]

[FIGURE REDACTED]

Results > Internal stiffness levels of engineered tumors vary with cell type:

“Although these observations of residual elasticity within spheroids is significantly higher than previously reported, these results are quite consistent with previous studies. Fresh metastatic tumor tissue sections probed by atomic force microscopy were found to be extremely heterogenous, containing stiffer regions (up to 16 kPa), compared to non-invasive tumors ⁷. The internal mechanical stress state of tissues can be very high when measured in live samples ⁵, and tissue sectioning is well-established to alter these stress states by releasing constraints and disrupting the active contractility of cells⁸ that would increase rigidity in non-linear biological materials ⁶. Our findings demonstrate the extent of this effect, further supporting the need for mechanical measurements without disrupting live tissue architecture. Furthermore, non-disruptive live techniques such as quantitative ultrasound have previously demonstrated internal tumor stiffness measurements up to 150 kPa³, albeit at considerably lower spatial resolutions. It is therefore likely that these measurements reflect highly focal and considerably larger rigidities that are “blurred” over a larger region. Hence, the measurements obtained with this novel technique are reasonable, and suggest that mechanical stimuli that promote invasion within tumors may be provided by small groups of cells within the spheroid population.”

4. It was claimed that the sensitivity of μ TAMs could be tuned (Fig. 2B simulation). However, the coupled expansion ratio and expanded stiffness of μ TAMs actually makes their sensitivity not so tunable in practice. For example, in Fig. 2C, the 9%N/0.6%B condition doesn't seem to be better suited for testing stiffer samples, which contradicts the idealized simulation.

We agree with this comment and have now softened our claims for tunability pending experimental validation of this idea. We do maintain that the system is tunable in principle, but also agree that we have not adequately demonstrated this concept. Hence, we have edited the text as follows:

Results > Sensor calibration and validation in engineered tissues: ... “Based on these experimental calibrations, we selected a formulation with the highest measurement sensitivity within the expected stiffness ranges for soft tissue. ”;

Discussion >, where we have removed claims of tuning the sensitivity, and discussed it as a possibility for future work: ...“Developing polymer engineering strategies to tune stored strain energy through independent manipulation of expansion ratio and sensor stiffness would enable precise manipulation of actuation force and stroke length, to better simulate mechanical interrogation by specific cell types and tune the sensors for various applications. “

5. In the study of animal tumor models, the result was interpreted as to suggest a link between highly localized mechanical properties within the tumor and disease progression towards metastasis (e.g. line 337-348). However, the development of those inoculated tumor models is very different from that of spontaneous tumors. Such claim can be misleading.

The 4T1 breast cancer mouse model used here is a well-established model to study the spontaneous transition from initial formation of a primary tumor through the metastatic cascade with invasion to distant sites. This model has been recognized as being similar to human mammary cancers⁹, and this process reliably occurs within three weeks (based on a starting injection of 500 cells in the mammary gland (Gregório et al (2016)¹⁰, refer to Table 2). We accelerated this process by injecting ~1250 cells⁹. In addition to the literature evidence and common usage of this model, we also experimentally confirmed both tumor formation and gradual to complete loss of key architectural features of the mammary tissues via histology over the experimental time course (Figure S7). Given the well-established use of this model to study tumor progression towards metastasis, and our supporting characterization, we believe that it is a reasonable starting point for us to investigate metastatic development.

We do recognize that despite the evidence presented of the 4T1 mouse as being a reasonable model of metastatic progression, inoculated and spontaneous tumors may differ in their initial architecture. We therefore agree that our claims should be limited to an association between metastatic state and observations of these highly localized stiff regions, and have carefully reviewed our manuscript to limit claims to these qualified statements. In addition, we have also recognized this limitation of standard models as a potential issue in the

Results > Long-term measurements of residual internal tumor stiffness evolution in animal models: ... “While it must be recognized that injection-based models may create tumor architectures that are different from spontaneously-arising tumors, these results do demonstrate that significant mechanical heterogeneity accompanies tumor progression *in vivo*, and correlatively suggests that highly-focal sites of rigidity may be sufficient to provide a mechanical stimulus for disease progression. “

6. The ex vivo μ TAM result shows that the average apparent moduli of breast tumor tissues decrease over time and are smaller than that of sham (Fig. 5E), which is inconsistent with previous reports using different mechanical testing methods (e.g. <https://doi.org/10.1038/s41551-016-0004>; <https://doi.org/10.1371/journal.pone.0193801>).

We respectfully disagree with these statements. We do not show in the figures or claim in the text that average tissue stiffness decreases over time: there is no statistically significant difference between the datasets (see Fig. 5E; $p = 0.9477$). We only claim that small sites of high stiffness exist within the tumor at later time points.

We also do not make any claims as to the overall stiffness of the tissue, but are instead measuring highly localized regions within the tumor tissue. While Voutouri and Stylianopoloulous (paper #2 cited by the reviewer) do show increases in the global excised tumor tissue stiffness, these measurements “lump together” all the stiffness variations that arise from macroscale tissue architecture, as well as the microscale tissue rigidity sensed by individual cells. Since we expect the tissues to be highly heterogeneous, the global stiffness measurements obtained in that work should not be compared with the measurements made in ours.

Similarly, we do not make any claims as to the stored elastic energy in our tissues. Nia et al., (paper #1 cited by the reviewer) develop an innovative technique to measure solid stresses, stored as elastic energy within tissues; and they establish in their paper that this is distinct from stiffness (see Nia et al.; Fig 2e-g). We do agree that this is confusing as stiffness and the solid stresses measured here have the same units, but they are distinct parameters: solid stress is the product of stiffness and strain. Furthermore, their measurements of bulk stiffness moduli through unconfined compression tests in thick tissue sections (Fig. 2e in their paper) also show no significant differences between primary and metastatic tumors, which is quite consistent with our data. Hence, our findings are generally consistent with the data reported in this work.

To address this important point and situate our findings in the context of other studies, we have now cited these two papers and provided further clarification regarding these differences in the Discussion section of the manuscript as follows:

Discussion > ... “ μ TAMs present some limitations that require careful consideration. First, the measurements obtained with this technique cannot be quantitatively compared with those obtained from more conventional approaches in materials that exhibit non-linear mechanical properties, such as viscoelastic tissues¹¹. Our current measurements of residual elasticity after creep can only be used to extract a stiffness modulus for materials that deform without a time-dependent component. Similarly, more advanced measurements that capture both mechanical stiffness and applied strain, such as internal solid stress⁸ also cannot be extracted.”

Reviewer #2:

1a. To demonstrate reliable measurements with this technique, the authors need to validate that μ TAMs can be used for measuring elastic moduli over the range encountered in their biological assays. The authors report stiffness values up to 250 kPa in later experiments, while they have only validated the measurement range against rheometry up to 20 kPa, with a fair bit of variation even at that stiffness.

We agree with this very important point, and have now extended our calibration and reproducibility measurements by measuring μ TAM expansion in stiff phantom tissues, extending the curve as seen below. The results are very consistent with our previous model and expectations. We have now updated our manuscript with these important extensions.

Figure 2C. Calibration curve of the 3% NiPAAM/ 0.2% bisacrylamide microgel formulation with the inclusion of μ TAM expansion within two super stiff polyacrylamide gels at 245 kPa and 270 kPa confirmed through rheology.

1b. Further, the authors should corroborate their local stiffness measurements using a conventional technique (microrheology, AFM, etc.) to provide more confidence in these measurements.

We would first like to clarify that we have already validated our local stiffness measurements against conventional measurement techniques, by performing shear rheometry on uniform polyacrylamide gels while calibrating this system (Fig. 2C,D). We carefully selected this method from the wide range of mechanical characterization techniques available, because most other techniques would not provide comparable measurements that match (1) the length scale, and (2) applied deformations of our target application, both of which significantly affect the measured mechanical properties of a hydrogel. These arguments are outlined as follows:

First, mechanical measurements in hydrogel biomaterials are well-established to change with the length scale of the measurement, as molecular structures, fibers, networks, and heterogenous networks have distinct mechanical properties¹². Techniques such as microrheology

probe hydrogels at the length scale of individual pores (~0.1 μm in polyacrylamide¹³. AFM measurements are on the length scale of 0.1 to 1 μm , depending on tip head size. In contrast, our application requires measurements at the length scale of 10s to 100s of microns, and while techniques such as nanoindentation or micropipette aspiration may provide more relevant length scales, they are limited to surface analysis, and cannot probe bulk tissue stiffness.

Second, the ‘stroke length’ of any applied deformation used in a measurement is of critical importance in obtaining mechanical properties, particularly for a hydrogel network¹⁴. Techniques such as microrheology exert virtually no deformation, and hence only provide a measure of pore size. While correlated to stiffness, this does not reflect stiffness of hydrogel network in response to deformation, as a cell would interrogate the matrix. The stroke length in AFM is on the order of microns, but is limited to surface measurements. Techniques such as optical tweezers and magnetic bead cytometry cannot generate sufficient force to create displacements of these magnitudes in stiff tissues. Magnetic twisting cytometry may be relevant, but measures shear deformation, rather than compressive modulus, which is also established to be different in hydrogel networks.

Hence, we chose to use a bulk measurement technique (shear rheometry) in polyacrylamide hydrogels because this material is well-established to be homogenous at the relevant length scales, and an accurate bulk measurement should closely reflect mechanical properties at the length scale of our sensors. Therefore, we believe that alternative techniques would not provide an apples-to-apples comparison for our studies, and we respectfully submit that alternative techniques would therefore not improve confidence in our measurements.

To address this comment and lay out the unique capabilities of our measurement technology, a condensed version of this discussion has been included in the manuscript introduction:

Introduction > “Conventional mechanical characterization techniques provide only a limited view of tissue stiffness, particularly at the “meso”-length scale of individual cells. Macroscale measurement tools such as tensional or shear rheometry cannot capture local stiffness variations around cells¹⁵, while high-resolution tools such as atomic force microscopy are ideally suited for sub-cellular nanoscale measurements, and are limited to measuring near-surface stiffness in two-dimensional or cut tissue sections. Although non-contact techniques such as ultrasound elastography or magnetic cytometry¹⁶⁻¹⁸ provide limited remote access to address these issues of scale, they cannot mimic a cell’s ability to interrogate the surrounding tissue by applied deformations with “stroke lengths” of 10s of microns^{19,20}

...Here, building upon recent materials-based strategies to generate local deformations within porous materials¹⁵, we introduce microscale temperature-actuated mechanosensors (μTAMs) that can measure a wide range of residual tissue elasticities within 3D biomaterials, at the length-scales of individual cells, in engineered tissues or animal models. μTAMs are spherical, thermoresponsive hydrogels that remain compact at tissue culture temperatures, but swell when cooled by a few degrees. By measuring the degree to which they expand, the residual elasticity after creep of the surrounding tissue can be inferred (Fig. 1A).”

2. There is a huge degree of variation in the reported stiffness values for measurements in cell spheroids. For example, in figure 4B, it appears that there is a difference in stiffness of three orders of magnitude over a few micrometer length. Since these drastic spatial differences in stiffness have not previously been observed, and since these stiffness values are much higher than those reported with other techniques, the authors should show using a standard measurement technique that these measurements are accurate.

We believe that our data presentation has accidentally misinformed the reviewer as to the experiment performed. While figure 4B does show stiffness readings as a function of radial depth into the spheroid, this is data that has been pooled from multiple spheroids each containing 1-2 μ TAM sensors. Hence we do not claim that measurements within a single spheroid change so drastically over a few microns, or that steep gradients exist within a tumor. Instead, our data demonstrates that we are taking measurements at various depths into the spheroid (Fig. 4B), and find that ~30% of readings in the MDA-MB-231 spheroids are unexpectedly high (Fig. 4C), and that this difference is statistically significant (Fig. 4D).

We hope that this clarification addresses the reviewers' surprise at these results. We do still agree that the high stiffness measurements are unusual, but as outlined in detail in our response to **Reviewer #1 (point 3c)**, we have validated our measurement system in phantom and manipulated spheroid tissues, and cannot compare our validated measurement system against other techniques because our system uniquely provides insight into intact 3D tissues at the cellular length-scale. In that response, we have also provided additional literature evidence, and some preliminary data demonstrating cellular organization around hydrogel beads within a spheroid, which we hope provides further confidence in these findings.

To address this misinterpretation, we have now clearly stated in the captions for Fig. 3 and 4 that measurement readings are pooled from multiple spheroid samples.

3. Many of the biological insights in this manuscript rely on inference from outlying data points. For example, in the abstract the authors state “we determine that small sites of unexpectedly high stiffness (>250 kPa) develop in invasive breast cancer spheroids, and in in vivo mouse model tumors.” It appears that the evidence for this statement regarding the mouse tumors is based on two measurement locations at day 14 and three at day 21 out of what seems to be dozens of data points. There does not appear to be any significant difference in the average stiffness along the time course of the experiment.

To provide focus for this comment, we reiterate that the comparative biological data presented in this manuscript in Figure 3 (revised version; comparison of spheroid fabrication techniques), and in Figure 4 (comparison of tumor cell types in spheroids) are not based on outlying data points, but on statistically significant comparisons between populations. Hence, this critique applies only to the data presented in Figure 5E (in vivo mouse model measurements).

The reviewer is of course, correct: there is no significant difference in our pooled measurements of random locations within the mouse tumor. Hence, we do not conclude that the average in vivo tumor stiffness changes over time, and have carefully reviewed our work to ensure that this is not claimed at any point in our manuscript. However, we can still claim the following:

- (a) these experiments show that “hot-spots” of mechanical stiffness do exist at later time points of cancer progression; and
- (b) that we were unable to detect these hot spots at earlier time points of tumor formation, or in sham control mice at 28 days.

Since statistical comparisons deal with populations by definition, they are not appropriate to describe the measurements being made here. **In this case, the lack of statistical significance between populations does not make these findings any less biologically important.** As an analogy, pathologists who identify a breast duct lesion are not asked to “average” that lesion (the outlier) over the entire tissue section (the population) to conclude that a cancer has formed. Similarly, we do not believe that a statistical averaging of these measurements is required to conclude that localized regions of high residual elasticity do exist within tumors at these length-scales, at sufficiently distinct stiffness levels that have otherwise been established to have a significant impact on cell fate and function²¹.

Furthermore, these findings of focal stiffening within tumors are consistent with the literature. Fibrotic disease mechanisms for example are well-established to be focally-driven by small groups of cells²². In cancer-specific studies, (Plodinec et al., *Nature Nanotechnology* 2012; previously cited in our manuscript), AFM measurements of tumor tissue sections demonstrate that a significant portion of even highly invasive cancer tissue exhibits a baseline stiffness of < 2 kPa with a relatively small percentage of areas with elevated stiffness (up to ~12 kPa). While the stiffness magnitudes are not comparable between our experiments, as the AFM technique

requires cutting tissue sections, which is well-known to release tissue stress⁸, the finding that variation exists still supports our current findings. These previous studies also contain no significance comparisons in their analysis of stiffness profiles, as the baseline readings would drown out measurements at the sites of focal stiffening.

To address this comment, we have:

- (1) carefully reviewed the entire manuscript to ensure that no wording can be misconstrued as a claim for differences between populations in these experiments; and
- (2) modified our Results section to clearly state the importance of these measurements:

Results > Long-term measurements of residual internal tumor stiffness evolution in animal models: ...
“More broadly, in diseases such as cancer where only a few aggressive cells are required to initiate metastasis, the ability afforded by μ TAMs to study highly localized mechanical microenvironments could ultimately provide an improved understanding of subpopulation-driven transitions between quiescent and malignant tumors.

4. The reported values for the stiffness of spheroids here differs greatly from prior reports. For example, Han et al (Nature Physics, 2019 doi: 10.1038/s41567-019-0680-8) showed that mammary spheroids were in the range of 25-100 Pa by microrheology and the Physical Sciences Oncology Centers Network (Sci. Rep. 3:1449, 2013) found the mammary cells had elastic moduli under 2 kPa by AFM indentation. Can the authors discuss the reasons for these discrepancies if they exist after more rigorous validation?

While multiple measurements of spheroid mechanical properties exist in the literature, the results reported here uniquely describe the residual elasticity within living, intact spheroids, at the length scale of individual cells, with an appreciable stroke length. Han et al. apply microrheology, a technique that applies a negligible deformation through thermal motion of nanoparticles, and provides a measure of pore size, which while correlated to stiffness, does not reflect the rigidity of a network of interconnected elements, which can be considerably different particularly for non-linear materials. While an AFM does generate a stroke length, these systems can only be applied to make surface measurements, either on small clusters or individual cells as in the paper cited by the reviewer, or on excised tissue sections⁷. These measurements cannot be performed within larger intact tissues, and cutting spheroids into sections releases stress⁸, changing the mechanical properties of the tissue. Hence, μ TAMS provide a unique measurement of internal tissue mechanics.

To address these comments, a new paragraph has been added to the discussion section to compare our measurements against existing results, including a discussion on how our measurements compare against other analysis techniques, as follows:

Discussion > “ μ TAMs provide an opportunity for high spatial-resolution measurements of residual elasticity after creep in a wide range of living, three-dimensional, tissues. Rather than measurements of global tumor mechanics, as has previously been reported^{23,24}, the tunable size of these sensors allows interrogation of mechanical properties at multiple length scales relevant to that of a biological cell, enabling an improved understanding of the local microenvironment that cells would experience. Furthermore, whereas sensitive analysis techniques as atomic force microscopy²⁵ and microrheology²⁶ may capture these spatial resolutions they do not approximate the stroke lengths generated by real cells, and hence measure mechanical properties of the material in a strain regime that may or may not be relevant to cellular mechanosensing and microenvironmental interrogation.”

5. Spheroids and soft tissues generally exhibit a significant degree of viscoelasticity. What is the timescale over which the spheroids/tissues are cooled? It seems probable that some of the stress induced by expansion of the microgel is dissipated by the viscous components of the tissue before measurements are made. Can the authors discuss how this might affect their measurements or potential avenues to overcome this issue?

The reviewer raises an excellent point. In our experiments, the spheroids are allowed to equilibrate for at least 30 minutes after the temperature setpoint is changed, prior to measuring the “expanded” μ TAM sizes. The following graph describes the characteristic temperature drop and μ TAM expansion over an extended period of time.

Figure S4C (new). μ TAM size change over time within T47D spheroids during cooling on our microscope stage. Temperature and size measurements were recorded every 5 minutes for an hour to ensure that 30 minutes was adequate time for μ TAM expansion to equilibrate, thereby ensuring residual elasticity around the μ TAM is measured.

Our temperature-controlled stage takes ~10 minutes to reach the critical transition temperature of 34 C, and an additional 20 minutes are allowed for the μ TAMs to equilibrate within the tissue. Within the 30 minute timeframe, viscous deformation of the spheroids/tissues could occur, and no further measurable changes in μ TAM size are observed. Hence (as pointed out by Reviewer 1, comment 1a), the μ TAMs are really measuring the “after creep” residual elasticity of the spheroid, and we have now taken care to clarify this concept throughout the manuscript.

Minor Comments

1) Can the authors explain why there is no observed increase in tumor stiffness along their experimental time course or why the sham injection is equally stiff as the tumors? Tumors that are stiffer than mammary fat pads and that stiffen over time have been observed by many others using multiple measurement modalities.

It is important to recognize that tumors vary based on model, and our experimental time-course may not be comparable with these existing studies. In terms of analysis methods, studies that have drawn this conclusion are typically based on global deformation of the tumor (example, work by Stylianopolous et al., refs 55 and 56 in the manuscript); measurement on sliced tissue specimens (example, Plodinec et al., ref 50 in the manuscript), or do not apply cell-like deformations to the tissue (example, Han et al., ref 58 in the manuscript), and these methodological differences may not allow comparisons between our studies, which report mechanics arising from cellular scale deformations in live, intact tumors. Furthermore, the factors contributing to global tissue stiffness include cellular stiffness, tumor cell and tissue architecture, internal and external stromal tissue architecture, and mechanics of the ECM present inside and on the tumor surface. In contrast to these studies, our system is applied here to uniquely measure stiffness within the tumoral regions only, in regions away from the surrounding fibrous stromal components, and at cellular length scales and stroke lengths.

It is therefore quite likely that previous reports of increasing tumor stiffness are related to factors at larger length scales where mechanics of the stroma can dominate internal tumor mechanics. For example, Fenner et al. (2014)²⁷ found that in mouse models, tumors can be softer, the same, or stiffer than the normal fat pad, but that this stiffness correlates with stromal collagen content rather than tumor architecture and cellularity. Similarly, Levental et al. (2009)²⁸ demonstrated that collagen crosslinking contributes to the overall stiffening around tumor tissue. Others have also shown that tumor regions adjacent to stiffer stroma remain soft⁷. Since we make measurements away from the stroma, it is hence not surprising that our results differ from conventional measurements of bulk tissue stiffness. Hence, we believe we are measuring a distinct parameter of internal tumor mechanics, which may be more relevant to mechanobiological activation of tumoral cells.

To address this issue, a concise summary of this discussion has been included in subsection “Long-term measurements of residual internal tumor stiffness evolution in animal models”

Results > Long-term measurements of residual internal tumor stiffness evolution in animal models: ...

“Previous studies have demonstrated that tumors can macroscopically soften, stiffen, or stay the same compared to adjacent normal tissue ⁴⁴, but whether this reflects macroscale tumor organization or microscale rigidity remains unclear. Furthermore, the surrounding stromal tissue stiffens with disease progression ⁴⁷, and it is challenging to eliminate those contributions when macroscopically probing live whole-tumor mechanics. Therefore, measuring stiffness within the living tumor itself, while restricting those measurements to length scales and stroke lengths relevant to individual cancer cells may yield new insights into tumor mechanobiology.”

2) It is not clear what the different colors represent in Fig. 1 C and D. Are these different individual microgels?

We apologize for the confusion. Each of the the different colours represents an individual microgel. Additional text has now been added to the Fig. 1 caption to clarify this.

3) It is not entirely clear what conclusions should be drawn from the two different spheroid generation strategies in Fig. 3.

We have now clarified the conclusions of this section based on the new data gathered for this revision. In summary, this data shows that the spheroid fabrication technique employed results in distinct internal mechanical stiffness levels. Since spheroid techniques are frequently used interchangeably, particularly between labs, these results demonstrate that subtle changes in method may result in distinct biological structures. We have clarified as follows:

Results > Internal spheroid mechanics differ with cell aggregation method:

“... When pooled together, spheroids formed through ATPS-induced aggregation exhibited significantly higher internal residual elasticity than those formed via micropocket-based aggregation (Fig. 3E).

Hence, conceptually similar fabrication methods produce spheroids with distinct internal tissue mechanics, and while the cause of these subtle differences remain uncertain, they may arise from small osmotic compressive pressures exerted by the dextran on the spheroids in the ATPS method ^{29,30}. Speculatively, these differences could spatially influence cell behaviour within the spheroid, which may contribute to explaining why biological findings vary considerably between research labs that use spheroids formed via slightly different methods³¹. In general however, these experiments establish the utility of μ TAMs in spatially characterizing internal mechanical rigidities that arise in 3D tissues.”

4) *The vast majority of the data points in Fig. 4B are indistinguishable from 0. Can the authors display this data differently, perhaps with a log scale y-axis? Similarly for Fig. 4D.*

We have updated these figures to have a log scale y-axis to better represent the full spread of data across several orders of magnitude.

5) *The y-axis label in Fig. 4C should be changed to frequency or something similar.*

Thank you, we have changed the y-axis label to “Normalized frequency”.

6) *Why are the fluorescent regions in Fig. 5B not spherical? How does this affect the measurements?*

Figure 5B shows the entire excised mammary gland, with a large number of fluorescent sensors deposited in the needle track. 3 days after injection (left panel), the μ TAMs are still in a dense bolus. As the tumor grows, the μ TAMs are distributed throughout the tissue (right panel). The macroscopic view provided only allows observations of clustered groups of μ TAMs as individual sensors are too small to see, but can be observed at higher magnifications (as in Fig. 5D). In our experiments, care was taken to exclude μ TAMs in close proximity to each other, to avoid neighboring sensors influencing the mechanical measurements. This important detail has now been included in the Methods section.

7) *The y axis should again be adjusted in Fig. S3 so that the data can be interpreted.*

Fig. S3B has been scaled to show the loss modulus data more clearly.

References used in this response document

1. Lee, W. *et al.* Dispersible hydrogel force sensors reveal patterns of solid mechanical stress in multicellular spheroid cultures. *Nat. Commun.* **10**, 144 (2019).
2. Dolega, M. E. *et al.* Cell-like pressure sensors reveal increase of mechanical stress towards the core of multicellular spheroids under compression. *Nat. Commun.* **8**, ncomms14056 (2017).
3. Chang, J. M. *et al.* Stiffness of tumours measured by shear-wave elastography correlated with subtypes of breast cancer. *Eur. Radiol.* **23**, 2450–2458 (2013).
4. Denis, M. *et al.* Correlating Tumor Stiffness with Immunohistochemical Subtypes of Breast Cancers: Prognostic Value of Comb-Push Ultrasound Shear Elastography for Differentiating Luminal Subtypes. *PLOS ONE* **11**, e0165003 (2016).
5. Liu, H. *et al.* In Situ Mechanical Characterization of the Cell Nucleus by Atomic Force Microscopy. *ACS Nano* **8**, 3821–3828 (2014).

6. Kasza, K. E. *et al.* The cell as a material. *Curr. Opin. Cell Biol.* **19**, 101–107 (2007).
7. Plodinec, M. *et al.* The nanomechanical signature of breast cancer. *Nat. Nanotechnol.* **7**, 757–765 (2012).
8. Nia, H. T. *et al.* Solid stress and elastic energy as measures of tumour mechanopathology. *Nat. Biomed. Eng.* **1**, 0004 (2017).
9. Pulaski, B. A. & Ostrand-Rosenberg, S. Mouse 4T1 Breast Tumor Model. *Curr. Protoc. Immunol.* **39**, 20.2.1–20.2.16 (2000).
10. Gregório, A. C. *et al.* Inoculated Cell Density as a Determinant Factor of the Growth Dynamics and Metastatic Efficiency of a Breast Cancer Murine Model. *PLOS ONE* **11**, e0165817 (2016).
11. Chaudhuri, O. *et al.* Hydrogels with tunable stress relaxation regulate stem cell fate and activity. *Nat. Mater.* **15**, 326–334 (2016).
12. Stamenović, D. & Coughlin, M. F. The Role of Prestress and Architecture of the Cytoskeleton and Deformability of Cytoskeletal Filaments in Mechanics of Adherent Cells: a Quantitative Analysis. *J. Theor. Biol.* **201**, 63–74 (1999).
13. Trappmann, B. & Chen, C. S. How cells sense extracellular matrix stiffness: a material's perspective. *Curr. Opin. Biotechnol.* **24**, 948–953 (2013).
14. Storm, C., Pastore, J. J., MacKintosh, F. C., Lubensky, T. C. & Janmey, P. A. Nonlinear elasticity in biological gels. *Nature* **435**, 191–194 (2005).
15. Proestaki, M., Ogren, A., Burkel, B. & Notbohm, J. Modulus of Fibrous Collagen at the Length Scale of a Cell. *Exp. Mech.* (2019) doi:10.1007/s11340-018-00453-4.
16. Wang, N. & Ingber, D. E. Probing transmembrane mechanical coupling and cytomechanics using magnetic twisting cytometry. *Biochem. Cell Biol.* **73**, 327–335 (1995).
17. Yamada, S., Wirtz, D. & Kuo, S. C. Mechanics of Living Cells Measured by Laser Tracking Microrheology. *Biophys. J.* **78**, 1736–1747 (2000).
18. Ashkin, A. Optical trapping and manipulation of neutral particles using lasers. *Proc. Natl. Acad. Sci.* **94**, 4853–4860 (1997).
19. Baker, B. M. *et al.* Cell-mediated fibre recruitment drives extracellular matrix mechanosensing in engineered fibrillar microenvironments. *Nat. Mater.* **14**, 1262–1268 (2015).
20. Buxboim, A., Rajagopal, K., Brown, A. E. X. & Discher, D. E. How deeply cells feel: methods for thin gels. *J. Phys. Condens. Matter Inst. Phys. J.* **22**, (2010).
21. Discher, D. E., Janmey, P. & Wang, Y. Tissue Cells Feel and Respond to the Stiffness of Their Substrate. *Science* **310**, 1139–1143 (2005).
22. Gabasa, M. *et al.* Epithelial contribution to the profibrotic stiff microenvironment and myofibroblast population in lung fibrosis. *Mol. Biol. Cell* **28**, 3741–3755 (2017).
23. Voutouri, C. & Stylianopoulos, T. Accumulation of mechanical forces in tumors is related to hyaluronan content and tissue stiffness. *PLOS ONE* **13**, e0193801 (2018).
24. Stylianopoulos, T. *et al.* Causes, consequences, and remedies for growth-induced solid stress in murine and human tumors. *Proc. Natl. Acad. Sci. U. S. A.* **109**, 15101–15108 (2012).
25. Agus, D. B. *et al.* A physical sciences network characterization of non-tumorigenic and metastatic cells. *Sci. Rep.* **3**, 1449 (2013).
26. Han, Y. L. *et al.* Cell swelling, softening and invasion in a three-dimensional breast cancer model. *Nat. Phys.* 1–8 (2019) doi:10.1038/s41567-019-0680-8.
27. Fenner, J. *et al.* Macroscopic Stiffness of Breast Tumors Predicts Metastasis. *Sci. Rep.* **4**, (2014).
28. Levental, K. R. *et al.* Matrix Crosslinking Forces Tumor Progression by Enhancing Integrin Signaling. *Cell* **139**, 891–906 (2009).
29. Montel, F. *et al.* Stress Clamp Experiments on Multicellular Tumor Spheroids. *Phys. Rev. Lett.* **107**, 188102 (2011).
30. Taloni, A., Ben Amar, M., Zapperi, S. & La Porta, C. A. M. The role of pressure in cancer growth. *Eur. Phys. J. Plus* **130**, 224 (2015).
31. Cisneros Castillo, L. R., Oancea, A.-D., Stüllein, C. & Régnier-Vigouroux, A. Evaluation of Consistency in Spheroid Invasion Assays. *Sci. Rep.* **6**, (2016).

Reviewers' Comments:

Reviewer #1:

Remarks to the Author:

The authors have provided additional results and discussions to address the previous comments by the reviewer. This work has improved substantially. The new interesting finding that spheroids fabricated using different methods have different residual elasticity is appreciated.

Reviewer #2:

Remarks to the Author:

The authors have done a nice job of clarifying and responding to previous comments. There are still a few areas of concern, listed below.

1) The previous review noted that this new technique should be validated using more conventional modalities. A similar comment was brought up by the other reviewer as well. While the authors have extended the range of the rheometry experiments with polyacrylamide gels, this doesn't fully assuage my concerns. The authors cite prior reports of AFM probing of tumor tissue showing elastic moduli up to 16 kPa. This is still well below the ranges reported using the technique developed in this paper ~ 300 kPa. While I appreciate the arguments that AFM probing may yield different results for a number of reasons the authors stated, it would still be more convincing to demonstrate that local stiffness values trend in the same direction as the μ TAM approach, even if the values are not identical.

2) Along the same lines, it is still not clear why the range of measured residual stiffness for MDA-MB-231 clusters spans from 100 Pa to > 200 kPa. The authors state in rebuttal "[we] cannot compare our validated measurement system against other techniques because our system uniquely provides insight into intact 3D tissues at the cellular length-scale." Thus, it is difficult to interpret these results, and will be difficult for others in the field to compare them to values obtained using more conventional techniques. I would recommend the authors attempt to compare these values with more conventional methods to at least demonstrate similar variation in spheroid mechanics.

3) The authors' comment about comparing only the mean value for Fig. 5 is well-taken. There are, however, methods for statistical comparisons of distributions meant to evaluate these types of data sets. Performing these comparisons of the distributions would provide confidence that the measurements reflect an important underlying biological difference between samples, and would strengthen the claims of the manuscript.

We thank the reviewer for once again engaging with our manuscript. We appreciate the comments raised, and have addressed each of them in the following response.

REVIEWER COMMENTS

1) The previous review noted that this new technique should be validated using more conventional modalities. A similar comment was brought up by the other reviewer as well. While the authors have extended the range of the rheometry experiments with polyacrylamide gels, this doesn't fully assuage my concerns. The authors cite prior reports of AFM probing of tumor tissue showing elastic moduli up to 16 kPa. This is still well below the ranges reported using the technique developed in this paper ~300 kPa. While I appreciate the arguments that AFM probing may yield different results for a number of reasons the authors stated, it would still be more convincing to demonstrate that local stiffness values trend in the same direction as the μ TAM approach, even if the values are not identical.

We believe that the core issue here is that our measurements of residual stiffness are unexpectedly high in MDA-MB-231 spheroids. While we address this underlying issue in detail in response to comment #2, we wanted to briefly note here that our arguments that AFM or other conventional techniques provide significantly altered values are neither speculative nor minor. Nia et al. (Ref 8 below) have clearly demonstrated that sectioning tumors for surface-based analyses such as AFM causes significant stress release through deformation. This is known to drastically change local tissue mechanics (see response to Q2). We maintain that there is no conventional technique that can provide measurements in these conditions (internal tissue residual elasticity; living and intact 3D tissues; 10s of microns spatial resolution; microns of stroke length). The unmet need for a simple technique to make these measurements is the reason we developed, validated, and applied μ TAMs to spheroids.

A literature search comparing conventional and novel mechanical characterization results for 3D cells, aggregates, and tumors has been included in the following section, along with new mechanical and tissue characterization results from our lab to further support this critical point.

2) Along the same lines, it is still not clear why the range of measured residual stiffness for MDA-MB-231 clusters spans from 100 Pa to > 200 kPa. The authors state in rebuttal "[we] cannot compare our validated measurement system against other techniques because our system uniquely provides insight into intact 3D tissues at the cellular length-scale." Thus, it is difficult to interpret these results, and will be difficult for others in the field to compare them to values obtained using more conventional techniques. I would recommend the authors attempt to compare these values with more conventional methods to at least demonstrate similar variation in spheroid mechanics.

Both comments 1 and 2 have common roots: our measurements of residual stiffness are unexpectedly high in MDA-MB-231 spheroids (as highlighted above). Since there seem to be no specific issues or suggestions regarding validation or utility of the μ TAM measurement technique itself, we will therefore focus this response on the core issue of "believability" of the high measurement values in spheroids.

In our previous response to the other reviewer (which was deemed sufficient), we included references to several papers demonstrating that our findings are both reasonable and plausible. We apologize for not originally recreating this directly in this section of the response, but have taken the opportunity to further expand this discussion by identifying additional stiffness measurements in similar tissues from the literature (summarized in table R2.1).

Table R2.1. Selected measurements of stiffness in relevant tissues.

#	Author	Stiffness value	Method	Tissue
1	Baker et. al. (2010)	$G' = 0.5$ to 2 Pa	Intracellular rheology	MCF7-10a (breast cell line)
2	Swaminathan et. al. (2011)	$E = 1$ to 6 Pa	Magnetic tweezers	ovarian cancer cell lines
3	Liu et. al. (2014)	$E = 3$ to 9 kPa	AFM	T24, RT4 (urothelial) cell lines
4	Calzedo-Martin et. al. (2016)	$E = 8$ to 20 kPa cell periphery and 20 to 700 kPa cell nucleus	AFM	MCF7-10A, MCF7, MDA-MB-231 (breast cancer cell lines)
5	Guillaume et. al. (2019)	$E = 2$ to 9 kPa	AFM	HCT116 spheroids (colon cancer cell line)
6	Jaiswal et. al. (2017)	$E = 0.23$ to 1.25 kPa	Microtweezers	BT474, T47D, MCF10A spheroids
7	Voutouri et. al. (2018)	$E = 50$ to 60 kPa	Compression testing	MCF10CA1a and 4T1 tumors
8	Nia et. al. (2016)	$E = 0.2$ to 20 kPa	Compression testing	Pancreatic and colorectal surgical resections
9	Denis et. al. (2016)	$E = 100$ to 147 kPa	Ultrasound elastography	Human breast tumours
10	Chang et. al. (2013)	$E = 136$ to 166 kPa	Ultrasound elastography	Human breast tumours
11	Khavari et al. (2019)	Internal stress ~ 1 to 4 MPa	Nuclei as pressure sensors	NIH 3T3 mouse fibroblast spheroids

Some common trends emerge from these publications. First, conventional mechanical characterization techniques (intracellular rheology, magnetic tweezers, AFM) that assess properties of cancer cell lines [1-4] typically yield values ranging from extremely soft ($G' = 0.5$ Pa; or $E \sim 1.5$ Pa) to reasonably stiff ($E = 20$ kPa), but have also shown that cancer cell nuclei specifically **can be as stiff as 700 kPa** [4], demonstrating that the cellular constituents of 3D tissues individually have the capacity to generate both low-rigidity and high-rigidity structures.

Second, intact cancer cell spheroids have been assessed by force-controlled AFM [5] and produce modulus values from 0.23 to 9 kPa. AFM techniques simply cannot probe deep within a spheroid, and are limited to surface analyses. Slicing tumors to expose internal surfaces for AFM has been shown to release internal stress [8], which significantly reduces a tissues' internal resistance to applied deformation (i.e., decreases apparent stiffness). AFM on sections of resected tumors demonstrates stiffness of 0.2 to 20 kPa [8] (similar to Plodinec et al., already cited in the manuscript), while compression testing of intact tumors shows stiffnesses up to 60 kPa [7]. The mismatch between these studies demonstrates that maintaining 3D tissue integrity is essential when characterizing internal tissue mechanics. Techniques such as AFM that disrupt tissues by sectioning or other means hence cannot be compared to the μ TAMs method developed in this work, and we cannot justify conducting these recommended experiments.

We then considered other conventional techniques that do not rely on tissue sectioning. We have previously discussed the impact of stroke length and force requirements in the previous response, and

will not repeat those arguments here, except to reiterate that this excludes optical / magnetic tweezers, and microrheology. Global compression of spheroids remains a possible alternative, and we therefore conducted preliminary compression-based measurements on spheroids using a commercially available cantilever-based force measurement probe (Microsquisher; CellScale Biomaterials Testing Ltd; Figure R2.1).

Figure R2.1. Microscale compression tests on intact spheroids. (A) Schematic of cantilever-based compression measurement apparatus. (B) Analysis of spheroid modulus based on tip displacements up to 10% of spheroid diameter. (C) Schematic of stress-shielding, in which preferential deformation of low-stiffness regions light blue in a heterogenous tissue prevents high-stiffness regions (orange) from deforming. Hence, high-stiffness regions contribute minimally to the measured stiffness, particularly when these regions are small and few.

Not surprisingly, we observe low global stiffnesses of 0.3 to 1.5 kPa, which is quite consistent with other studies of cancer spheroids (microtweezers; [6]). These global measurements simply cannot be compared with local measurements, because stress shielding within the spheroid would mask any regions of high stiffness. In stress shielding (Fig. R2.1 C), soft regions “absorb” the applied global deformations around any embedded stiff zones, masking the presence of small, stiff regions within the heterogenous tissue. For microscale experimental characterization of this well-known phenomenon, please see Moraes et al., *Biomaterials* 2010 [12] amongst many other studies. Hence, any global or bulk measurements of stiffness would drastically under-report small regions of high stiffness within the tissue. Since these global measurements do not provide an apples-to-apples comparison with the μTAMs system, we have not included these experiments in our revised manuscript.

Since the only valid comparisons with μTAMs is to measure stiffness within intact, living tissues, we review the literature on ultrasound shear wave elastography imaging [9-10]. These specialized techniques have spatial resolutions of approximately 150-200 μm, and show “hot spots” of stiffness up to 166 kPa in human tumors. Intriguingly, these studies could be capturing a much smaller and stiffer region that is “blurred” or averaged over a larger area, Hence, our measurements of > 250 kPa over 10s of microns are quite reasonable, and ultrasound elastography supports our findings that very high stiffnesses can arise within living, intact, cell-dense tumor material.

It is important to recognize that the tumors in the ultrasound studies arise over several weeks-to-months, whereas our spheroids are formed over a few days. Whether hot spots of high stiffness can arise after just a few days in culture remains an open question. While we do believe that our μTAM

technique allows us to conclude that this can happen, other novel high-resolution techniques also demonstrate that this is possible. By characterizing the mechanical properties and deformations of nuclei, Khaveri et al. [11] recently mapped local internal tissues stresses at the individual cellular level in two-day old fibroblast spheroids, and found internal **compressive stresses up to 4 MPa at this length-scale**. Although not technically a “modulus”, these stresses must correlate with a materials’ resistance to deformation (applied in opposing directions, as is the case with μ TAMs). These studies further support our findings that high-spatial resolution measurements within newly-formed, living three-dimensional spheroids can yield extremely high measurements of rigidity.

Given that evidence exists for high stiffness within newly-formed spheroids, we have also wondered what might create these zones. While this is beyond the scope of the current paper, we speculate that active cell mechanical traction forces and local tissue architecture in living tissues may be responsible. We have obtained preliminary evidence demonstrating that cells form polarized regions around embedded soft hydrogel particles, and that this organization is lost away from the particle (data previously shown in Response 1, to Reviewer 1 comments; Fig. R2.2 below).

[FIGURE REDACTED]

[REDACTED]

Although this literature review and preliminary experiments are certainly not a conclusive re-validation of our findings, we do hope that they at least demonstrate to the reviewer that our measurements are reasonable and consistent with other studies in the field, particularly those in live, intact, 3D tissues. We do not see any other conventional way to obtain similar measurements of stiffness at this length-scale. Given the available literature evidence for non-conventional measures in intact spheroids, the known limitations of conventional techniques, and the experiments we have already performed to validate the μ TAMs measurement technique itself, we respectfully disagree that these recommended experiments would add scientific value to the present work. Since these references were included with the previous revision, we have not modified the manuscript based on this feedback.

3) *The authors' comment about comparing only the mean value for Fig. 5 is well-taken. There are, however, methods for statistical comparisons of distributions meant to evaluate these types of data sets. Performing these comparisons of the distributions would provide confidence that the measurements reflect an important underlying biological difference between samples, and would strengthen the claims of the manuscript.*

We thank the reviewer for this suggestion – we had not previously considered comparing the distributions using statistical methods. To help us apply the appropriate techniques to the *in vivo* data presented in Figure 5, we consulted with a colleague (Dr. Clement Ma, Senior Biostatistician, Harvard Medical School) to determine the most appropriate methods to support our current claims, and extend them to statistically compare distributions.

We first asked whether descriptive statistics might be helpful to support our current claims that these “hot spots” are consistent features in our study. Descriptive statistics allows us to state the probability that our outlying data points would have been observed in the sham control experiment (log-transformed to obtain a normal distribution, confirmed via a Shapiro-Wilks test, p-value of 0.400). We obtained z-score values of 2.16 to 2.81 for the three largest data points in the 3-week old tumors. This means that there is less than a 1.5% probability that any of these extreme values would be observed in the sham control experiment (or that this experiment would have to be repeated ~65 times to find just one of these large values in the sham control dataset). For the largest measurement, this probability was less than 0.25%. A similar analysis of our *in vitro* experiments comparing T47D and MDA-MB-231 cells (Figure 4; loosely analogous to early- and late- time points in our mouse model) showed even more extreme z-scores up to 6.93 (probability is $< 1 \times 10^{-9}$ %). Since these extreme values appear in both the well-defined *in vitro* models and in the *in vivo* mice models, this supports our original conclusions that these “outlying” data values are consistent and expected features in this study.

As suggested by the reviewer, we then asked whether we could extend our claims to comparing the measurement distributions as being significant features of tumor progression. Since data for each of our time-points was obtained from an independent (sacrificed) mouse, and the data is not normally distributed, we used a nonparametric Mann-Whitney test to compare the distribution of ranked data, and found a statistically significant difference ($p = 0.022$) in measurement distributions between Week 1 and Week 3 of tumor progression. Comparisons with the sham control (healed adipose tissue) are not relevant to draw conclusions about tumor progression. While these tests do not conclude that tissues change in mean residual elasticity, we can now confirm that the distribution (i.e, the heterogeneity of measurement in the tissues) changes significantly during tumor progression.

To incorporate these new analyses into the manuscript we have:

(1) Included significance information in Figure 5E, and updated the caption as follows:

Figure 5... (E) Comparison of residual elasticity within tumors indicates an increasing number of high-rigidity measurements as the cancer progresses towards metastasis, and a significant difference in measurement distributions between Day 7 and Day 21 of tumor progression (* $p = 0.022$ by non-parametric Mann-Whitney test to compare the distribution of ranks between groups). Box plots indicate the median and first to third quartile, whiskers span the range.

(2) Modified our Results section to clearly state that the differences in distribution are statistically significant

Results > Long-term measurements of internal tumor rigidity in animal models

Although the mean measurements of internal tumor rigidity did not change significantly over 21 days, we observed a significantly different distribution of measurements as the tumor progressed from day 7 to day 21 (Fig. 5E; * $p = 0.022$), with some sites stiffening to between 25 and 50 kPa. The probability of measuring these high values in the sham control experiment are between 0.25% and 1.5%, based on descriptive z-score statistics. Certain regions within the mouse tumors were therefore much more rigid than the overall tumour by day 14, which matches both our findings that local mechanical heterogeneity increases in invasive engineered tumors (Fig. 4), and observations of increasing architectural heterogeneity and stromal organization as the tumor overtakes normal tissue (Fig. 5C). ... these results do demonstrate that significant differences in mechanical heterogeneity accompany tumor progression in vivo.

(3) Included a description of these statistical methods in the Supplementary Methods section of the manuscript.

Statistical analysis

Z-scores were used to assess probability of obtaining measurements compared to a control population. For non-normal distributions, log transformations were used to first obtain normal distributions, which were confirmed via Shapiro-Wilks tests. Comparative data analysis of populations was performed without pre-specifying a required effect size. Datasets that were normally distributed, with similar variances between compared groups were analyzed using unpaired t-tests or two-way ANOVA to test for significance which was set at $\alpha = 0.05$. Post-hoc pairwise comparisons were conducted using the Bonferroni method. Datasets that were not normally distributed were analyzed using the nonparametric Mann-Whitney test to compare the distribution of ranks between two groups, with significance values set at $\alpha = 0.05$. All statistical analyses were performed using GraphPad Prism v8.0.2 (San Diego, CA).

(4) Indicated Dr. Ma's contribution in the Acknowledgements section of the manuscript

We once again thank the Reviewer for engaging with our work, and appreciate the thoughtful comments, suggestions, and opportunities for discussion.

References

1. Baker, E. L., Lu, J., Yu, D., Bonnecaze, R. T. & Zaman, M. H. Cancer Cell Stiffness: Integrated Roles of Three-Dimensional Matrix Stiffness and Transforming Potential. *Biophys. J.* 99, 2048–2057 (2010).
2. Swaminathan, V. *et al.* Mechanical Stiffness Grades Metastatic Potential in Patient Tumor Cells and in Cancer Cell Lines. *Cancer Res.* 71, 5075–5080 (2011).
3. Liu, H. *et al.* In Situ Mechanical Characterization of the Cell Nucleus by Atomic Force Microscopy. *ACS Nano* 8, 3821–3828 (2014).
4. Calzado-Martín, A., Encinar, M., Tamayo, J., Calleja, M. & San Paulo, A. Effect of Actin Organization on the Stiffness of Living Breast Cancer Cells Revealed by Peak-Force Modulation Atomic Force Microscopy. *ACS Nano* 10, 3365–3374 (2016).
5. Guillaume, L. *et al.* Characterization of the physical properties of tumor-derived spheroids reveals critical insights for pre-clinical studies. *Sci. Rep.* 9, 6597 (2019).
6. Jaiswal, D. *et al.* Stiffness analysis of 3D spheroids using microweepers. *PLoS ONE* 12, (2017).
7. Voutouri, C. & Stylianopoulos, T. Accumulation of mechanical forces in tumors is related to hyaluronan content and tissue stiffness. *PLOS ONE* 13, e0193801 (2018).
8. Nia, H. T. *et al.* Solid stress and elastic energy as measures of tumour mechanopathology. *Nat. Biomed. Eng.* 1, (2016).
9. Denis, M. *et al.* Correlating Tumor Stiffness with Immunohistochemical Subtypes of Breast Cancers: Prognostic Value of Comb-Push Ultrasound Shear Elastography for Differentiating Luminal Subtypes. *PLOS ONE* 11, e0165003 (2016).
10. Chang, J. M. *et al.* Stiffness of tumours measured by shear-wave elastography correlated with subtypes of breast cancer. *Eur. Radiol.* 23, 2450–2458 (2013).
11. Khavari, A. & Ehrlicher, A. J. Nuclei deformation reveals pressure distributions in 3D cell clusters. *PLOS ONE* 14, e0221753 (2019).
12. Moraes, C., Wang, G., Sun, Y., Simmons, C.A. “A microfabricated platform for high throughput unconfined compression of micropatterned biomaterial arrays”, *Biomaterials* 31(3) pp. 557-84 (2010)

Reviewers' Comments:

Reviewer #2:

Remarks to the Author:

In reference to my first and second comments, I find the authors' arguments regarding limitations of other techniques persuasive, and recognize the difficulty in performing these validation experiments, particularly in light of the current restrictions on in-lab experiments. I appreciate the thorough response and summary of prior results from different measurement techniques. Based on the literature review, unpublished data presented here, and that the other reviewer is satisfied, I am willing to accept without the additional validation experiments.

With the added statistical analyses in Fig. 5, the authors have addressed my comment.